# Humans optimally anticipate and compensate for an uneven step during walking

Osman Darici*, Arthur D Kuo

Faculty of Kinesiology, University of Calgary, Calgary, Canada

**Abstract** The simple task of walking up a sidewalk curb is actually a dynamic prediction task. The curb is a disturbance that could cause a loss of momentum if not anticipated and compensated for. It might be possible to adjust momentum sufficiently to ensure undisturbed time of arrival, but there are infinite possible ways to do so. Much of steady, level gait is determined by energy economy, which should be at least as important with terrain disturbances. It is, however, unknown whether economy also governs walking up a curb, and whether anticipation helps. Here, we show that humans compensate with an anticipatory pattern of forward speed adjustments, predicted by a criterion of minimizing mechanical energy input. The strategy is mechanistically predicted by optimal control for a simple model of bipedal walking dynamics, with each leg's push-off work as input. Optimization predicts a triphasic trajectory of speed (and thus momentum) adjustments, including an anticipatory phase. In experiment, human subjects ascend an artificial curb with the predicted triphasic trajectory, which approximately conserves overall walking speed relative to undisturbed flat ground. The trajectory involves speeding up in a few steps before the curb, losing considerable momentum from ascending it, and then regaining speed in a few steps thereafter. Descending the curb entails a nearly opposite, but still anticipatory, speed fluctuation trajectory, in agreement with model predictions that speed fluctuation amplitudes should scale linearly with curb height. The fluctuation amplitudes also decrease slightly with faster average speeds, also as predicted by model. Humans can reason about the dynamics of walking to plan anticipatory and economical control, even with a sidewalk curb in the way.

*For correspondence:
osman.darici1@ucalgary.ca

Competing interest: The authors declare that no competing interests exist.

## Editor's evaluation

The study combines theoretical and experimental approaches to probe the laws governing strategy for coping with the control of stepping on uneven terrain. Congruent results in anticipatory and reactive adjustments indicate that a simple strategy based on the conservation of energy may be expressed within neural control pathways for locomotion.

## Introduction

There are indeterminate control choices to be made during walking, not least when steady gait is interrupted by a surface perturbation such as a sidewalk curb (*Figure 1a*). If no compensation is performed, an upward perturbation might cause a sudden reduction of speed, and an overall loss of time compared to steady gait (*Figure 1b*). Other possibilities include nonanticipatory reaction occurring only after the perturbation (*Figure 1c*), or tight regulation of step timing to immediately restore momentum (*Figure 1d*). But it may be helpful to plan and act ahead with anticipatory adjustments. For example, a ball, once sighted, may be caught by predicting its dynamics and planning and executing an interception course. A sidewalk curb may similarly be anticipated ahead of time, albeit with most

**Figure 1.** Hypothetical ways to walking over a single upward step (Up-step). (**a**) Humans walk with the body center of mass (COM) moving up and down atop the stance leg behaving like an inverted pendulum. Momentum is disrupted by the Up-step if not for compensatory response, plotted as a time-varying trajectory of the COM's forward speed $v_i$, discretely sampled once per step . Hypothetical responses could include (**b**) no compensation, (**c**) a nonanticipatory reactive compensation, and (**d**) tight time regulation. (**b**) No compensation means the same push-off actuation is performed regardless of Up-step, resulting in a transient loss of speed (speed vs. time, top) and time (cumulative time gain vs. time, bottom) relative to nominal level gait. (**c**) Nonanticipatory compensation occurs reactively after the perturbation (vertical dotted line aligned to Up-step) to quickly regain speed, perhaps to avoid loss of time. (**d**) Tight time regulation means applying control to keep each step's timing as constant as possible to reject the disturbance. These responses could be less economical than an optimal control including anticipatory speed adjustments before and after the perturbation. In plots, speed $v_i$ is sampled discretely at midstance instant (dot symbols) and plotted vs. time. Cumulative time gain is illustrated as time gained or lost compared to steady-level gait at nominal speed, as a function of elapsed time during a walking bout. The final value indicates the eventual time gain (or loss, if negative) over the entire distance. In the corresponding experiment, human subjects walked in a walkway (30 m long) with level ground ora single Up- or Down-step (height $b$ = 7.5 cm) at midpoint.

of the dynamics within the person rather than the curb, and with a less clearly defined objective. This raises the question of whether humans anticipate the curb, and what objective criteria govern its interception, perhaps with a single, optimal response. The seemingly simple task of dealing with an uneven step may yield insight on whether humans perform predictive, dynamical planning while they walk.

Both feedback and anticipatory control could contribute to walking. Feedback refers to reactive control driven by sensory feedback of the body's dynamical state, for example from vestibular and somatosensors to control standing balance (*Horak et al., 1990*; *Kuo, 1995*; *Park et al., 2004*). Such feedback also appears important for balance during walking, for example to adjust foot placement each step (*Bauby and Kuo, 2000*; *O'Connor and Kuo, 2009*; *Wang and Srinivasan, 2014*). In contrast, anticipatory control actions take place before the body state has been affected, perhaps using vision to predict an upcoming perturbation or obstacle. For example, humans clearly anticipate

and plan for the body's future location, for example to negotiate around obstacles or through door-ways (*Arechavaleta et al., 2008*; *Brown et al., 2021*; *Patla, 1998*). In particular, foot placement and motion are planned, with the help of vision, to avoid or step over upcoming obstacles (*Patla, 1998*; *Patla and Rietdyk, 1993*), and to land on foot targets (*Drew et al., 2008*; *Matthis and Fajen, 2013*). Although much of this planning has been described in terms of kinematics, it may also include intersegmental dynamics, which might be regulated to clear obstacles with reduced mechanical work (*Patla and Prentice, 1995*). There may, however, also be advantages to planning other dynamical states such as speed or momentum during locomotion, for example when one speeds up before jumping over a puddle. Indeed, runners do load the leg differently just before a drop (*Müller et al., 2012*), perhaps as a way to regulate momentum. The negotiation of uneven terrain might similarly benefit from anticipatory planning of speed or momentum.

Any systematic control strategy, regardless whether anticipatory or feedback, should also be driven by objective criteria to select among infinite options. For locomotion, the criterion of metabolic energy economy determines features such as the preferred step length and step width during steady walking (*Donelan et al., 2001*; *Zarrugh et al., 1974*), as governed by the pendulum-like dynamics of the legs (*Kuo et al., 2005*). Energetic costs are also greater on uneven terrain (*Kowalsky et al., 2021*; *Pandolf et al., 1977*; *Voloshina et al., 2013*) and under transient conditions (*Brown et al., 2021*). However, it is unknown whether the economy preferences of steady walking apply to uneven terrain as well. We previously explored this question theoretically with a simple model of human walking (*Kuo, 2002*), for the task of negotiating a single uneven step (termed Up- or Down-step) during otherwise steady walking (*Darici et al., 2020*). We used optimal control to determine the most economical strategy to regain time and momentum lost from the disturbance. The objective was to minimize a crude indi-cator of energy expenditure, the mechanical work performed in the step-to-step transition from one pendulum-like stance leg to the next (*Donelan et al., 2002*; *Kuo et al., 2005*). This yielded a strategy for negotiating an Up-step by modulating forward momentum and speed over multiple steps, starting with a speed-up before the perturbation, and continuing the modulation for several steps after the perturbation. The model predicted a substantial economic advantage to the anticipatory speed-up, compared to post hoc compensation alone. The optimal control was systematically scalable, with greater amplitude of speed fluctuations for larger step heights, and an almost opposite amplitude for negotiating a Down-step (with negative height change). This suggests that humans might also benefit from such an anticipatory strategy, made determinate by optimization.

The purpose of the present study was to test whether humans negotiate a step height disturbance with an anticipatory compensation strategy as predicted by dynamic optimization. The compensation is manifested as a trajectory of speed fluctuations both before and after the disturbance, to reduce energy expenditure and avoid loss of time from the disturbance. We tested whether humans use a compensatory pattern similar to the predicted optimum, including anticipatory fluctuations before the step is physically encountered. This is contrasted against alternative strategies such as reacting only after the disturbance, or not reacting at all. We also tested whether the human pattern varies system-atically, as predicted by model (*Figure 2*), by scaling in amplitude and time for Up- vs. Down-steps and different walking speeds. This may reveal whether such a compensation strategy may be generalized for different nominal walking conditions. The results may reveal whether humans reason about their walking dynamics to perform predictive planning on uneven terrain.

## Model predictions

The model yielded two main predictions to be tested experimentally. First, it predicted an antici-patory compensation strategy, in the form of speed fluctuations before and after the disturbance (*Figure 3*). Considerable momentum and time would be lost to an Up-step if not for compensation. The predicted optimal compensation to avoid a loss of overall time is to speed-up in advance of the Up-step (thereby reducing the loss of momentum and time atop it), and then regain momentum afterwards (*Figure 3a*; *Darici et al., 2020*). The strategy depends systematically on the disturbance amplitude, where a Down-step is represented by negative amplitude. For stepping down (*Figure 3b*), the optimal strategy is therefore almost exactly opposite the Up-step strategy: slow-down in advance, gain speed and time atop the Down-step, and then slow-down again toward nominal speed. This is manifested as twofold symmetry, meaning the optimal speed fluctuations are opposite in time (reflecting about the vertical axis at time 0) and in speed (reflecting about the mean speed). These

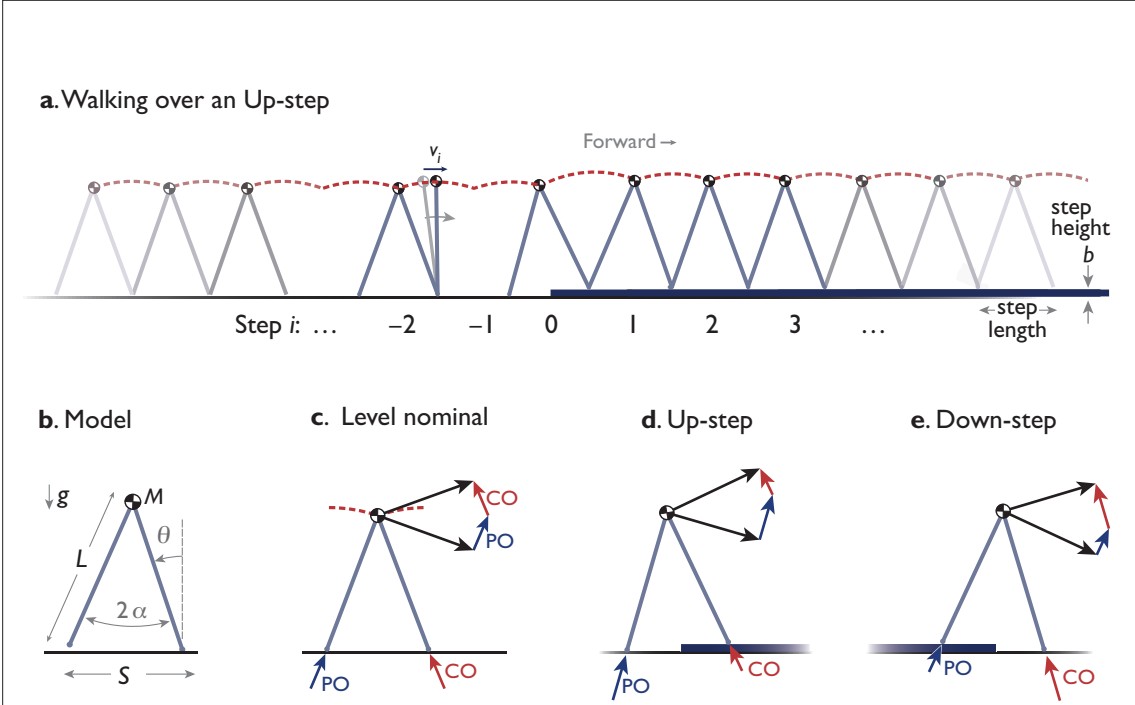

**Figure 2.** Model of dynamic walking over a single Up- (or Down-) step. (**a**) Model behaves like an inverted pendulum. Momentum and speed fluctuate in each step (numbered ), and are particularly disrupted by an uneven step (at $i = 0$). (**b**) Dynamic walking model has a point mass $M$ at pelvis, supported by an inverted pendulum stance leg (massless, length $L$, gravitational acceleration $g$, fixed or constrained step length $S$ and inter-leg angle $2\alpha$). (**c**) Level nominal walking has discrete step-to-step transition where COM velocity (dark arrow) is redirected from forward-and-downward to forward-and-upward by active, impulsive trailing leg push-off (PO), immediately followed by an inelastic, impulsive, leading leg collision (CO). Both PO and CO are directed along the corresponding leg. (**d, e**) The model walks Up or Down a step by modulating the sequence of discrete push-offs surrounding and including the uneven step. The model's compensatory response is therefore summarized as a time-varying trajectory of speeds $v_i$, discretely sampled once per step .

strategies are executed through modulation in push-off work. As a result, time (*Figure 3c*) is first gained prior to the Up-step (and lost prior to Down-step, *Figure 3d*), such that the cumulative time gain eventually reaches zero, meaning that the model has not lost time compared to level walking.

The second main prediction was that a single optimal strategy, in terms of speed fluctuations, is scalable for practically any combination of overall walking speed and step length (termed self-similarity, *Figure 3d*). A basic pattern for optimal speed fluctuations retains approximately the same shape across different overall walking speeds, fixed step lengths, or even step length changing according to the human preferred step length relationship (*Figure 3d*). In addition, the amplitude of that basic pattern scales inversely with speed, meaning slightly smaller fluctuations for faster speeds (compare slower to faster speeds in *Figure 3d*). This is because a step of fixed height (and thus gravitational potential energy) has a relatively smaller effect on the greater kinetic energy (increasing with square of speed) of faster walking. In addition, the timing scales such that the optimal strategy would have similar shape with shorter step lengths, except stretched slightly longer in time. Thus, we expect a single basic pattern, treated as a sequence of discrete speed fluctuations (*Figure 3a*) to predict optimal responses that are scalable to an individual's average speed, step length, or the preferred speed and step length relationship. The pattern should be scaled to slightly smaller amplitude for faster average speeds, and to negative amplitude for Down-steps as opposed to Up-steps.

These predictions may be contrasted with several alternative hypothetical strategies (detailed by *Darici et al., 2020*). One is that humans might not compensate for an Up-step, and simply apply the same nominal effort as for level ground, albeit with a loss of time relative to nominal (*Figure 1b*). Another is that humans may not anticipate but react only after the disturbance, which could regain lost time but with a greater speed-up (*Figure 1c*). Yet another would be to tightly regulate step timing to nominal with each step (*Figure 1d*), even atop the disturbance, and thus keep speed fairly steady.

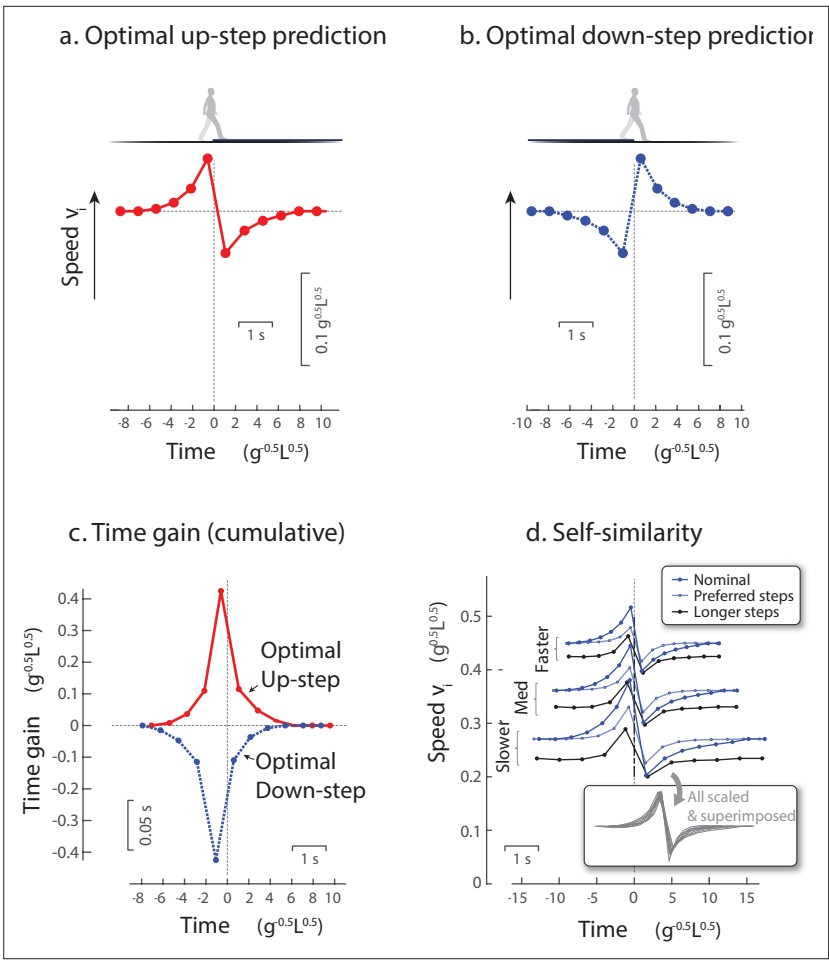

**Figure 3.** Model predictions for walking over an Up-step or Down-step with minimum work. (**a**) Optimal walking speed fluctuations vs. time, for Up-step compensation that minimizes push-off work while avoiding loss of time. Model anticipates the perturbation with a tri-phasic adjustment: Speed up ahead of time, then lose momentum atop the perturbation, and then regain speed thereafter. (**b**) Optimal speed fluctuations for Down-step compensation (blue symbols) is also tri-phasic, and nearly opposite in sign to Up-step: Slow down in advance, gain momentum, then slow down again (**c**) Cumulative time gained for Up- and Down-step compensations, ending with zero cumulative time loss. (**d**) Self-similarity of Up-step compensations shows that a similarly-shaped discrete compensation pattern can apply to different walking conditions with appropriate scaling. Example trajectories are shown for three different nominal speeds (slower, medium, faster), a longer fixed step length at each nominal speed (longer steps), and step length increasing with instantaneous speed according to human preferred step length (preferred steps). The trajectories are also scaled and superimposed (see inset) to illustrate a single self-similar pattern for all parameter choices. For model predictions (a-c), nominal conditions are equivalent to a human walking at 1.5 m/s with a fixed step length, and a 7.5 cm Up-step. Plots show normalized units (nominal mid-stance velocity $V = 0.44 g^{0.5} L^{0.5}$, $S = 0.79 L$, $b = 0.075 L$) and human scale. Predictions are described in detail by (*Darici et al., 2020*).

According to the model, these strategies all differ in the amount of work performed, 0.489 MgL (about 10% more than nominal walking) for the optimal strategy, 0.519 MgL (17% more) for nonanticipatory reactive control, and 0.513 MgL (15% more) for tight time regulation. It would of course be most economical not to compensate at all. But if time is to be regained, there are infinite ways to do so, and only one that is most economical.

The model therefore makes testable predictions regarding human compensation for an uneven step. We hypothesize that step-to-step transitions are costly enough to merit anticipation and compensation for uneven steps. There are surely other energetic costs for walking, such as for adjusting step duration and length, or for controlling other degrees of freedom. It is, however, not necessary to model such features, if step-to-step transitions are costly enough to predict substantial and nontrivial

compensations. We test what strategy humans perform by examining the distinct speed (and thus momentum) fluctuation patterns (*Figure 3*, top).

## Results

Prior to the main hypothesis tests, we first report some basic measures of overall walking speeds and variability, as a basis for comparing speed fluctuations (Speed, *Figure 4*). For the central (8.5 m) segment of the walkway, the overall average self-selected speed for control trials (level walking) was 1.38 ± 0.10 m/s on level ground (mean ± standard deviation [SD across subjects]). Each individual typically had a small amount of variation in self-selected speed between trials, with about 5% c.v. (coefficient of variation) across control trials. Subjects were thus fairly consistent in their own walking speed, despite receiving no feedback regarding walking durations or speeds. During level Control walking (*Figure 4*, top row), the speed fluctuations were small in magnitude and largely noise-like, with variability 0.031 ± 0.007 m/s (root-mean-square variation within trial, reported as mean ± SD across subjects), or about 2.2% c.v. These fluctuations exhibited a small amount of correlation between subjects, with correlation coefficient $\rho = 0.47 \pm 0.31$ (p = 2.5e−04), as demonstrated by correlating each individual's average Control trial against the average Control across all subjects. This suggests a degree of unexpected, nonrandom behavior shared between subjects, of relatively small amplitude of about 0.014 m/s, about 17% of the non-Control amplitudes.

We next examine the human walking speed fluctuations in response to Up- and Down-steps. There were four main findings, comparing the responses across subjects in comparison to model, and in comparison to experimental instructions. The first question was whether the Up- and Down-step responses exhibited consistent, nonrandom behavior across subjects, to determine whether speed fluctuations were noisy or deterministic. The second was to test the primary hypothesis, that human speed fluctuations have a deterministic pattern as predicted by a model minimizing mechanical work. The third was to test the related expectation that speed fluctuation amplitudes should scaled with step height and overall walking speed, as also predicted by model. The final test was to quantify the consistency of overall walking durations for each subject, as an indicator of compliance to experimental instructions. Along with their speed fluctuations, subjects also exhibited fluctuations in step length and duration. These factors were not included in model and therefore not tested, but the trends are summarized in Appendix 2.

### Humans produced triphasic Up- and Down-step compensatory speed fluctuations

There was also a clear pattern in compensations for an uneven step, with consistent fluctuations in walking speed across trials and across subjects (Speed, *Figure 4*). The fluctuations within these trials were greater than those of Control, about 3.0% and 3.4% c.v. for Up- and Down-steps (*Figure 4*, middle and bottom), respectively. The compensation strategies, in terms of walking speed trajectory over time, appeared qualitatively similar between multiple trials for an individual (*Figure 4*, left column), and between different individuals (*Figure 4*, middle column), to yield a single representative trajectory for all Up-step compensations (*Figure 4* right column).

The basic speed response could be summarized in terms of a triphasic pattern centered about the Up-step: (1) speed up in the two steps prior, (2) then lose speed during the two steps onto the Up-step and immediately thereafter, and (3) then regain speed over the following one or two steps. The peak speed just prior to the Up-step ($i = -1$) was about 5.7% greater than average speed, and the minimum after the Up-step ($i = 1$) was about 3.4% slower. Similar observations were the case for Down-step compensations (*Figure 4*, bottom row), except with fluctuations in nearly the opposite direction, in a basic pattern of about 2.0% slow-down prior to the Down-step ($i = -1$), and 3.6% and 3.8% faster after ($i = 1$, $i = 2$ respectively), followed by a brief slow-down to nominal. The timing of this pattern was slightly different from Up-step, with the initial slow-down being clearest for only one step immediately before the Down-step ($i = -1$), then speed-up occurring for about two to three steps, and the return to normal walking in only about one step. The anticipatory compensation thus begins only a few steps before the perturbation, and is accompanied by several steps of response after the perturbation.

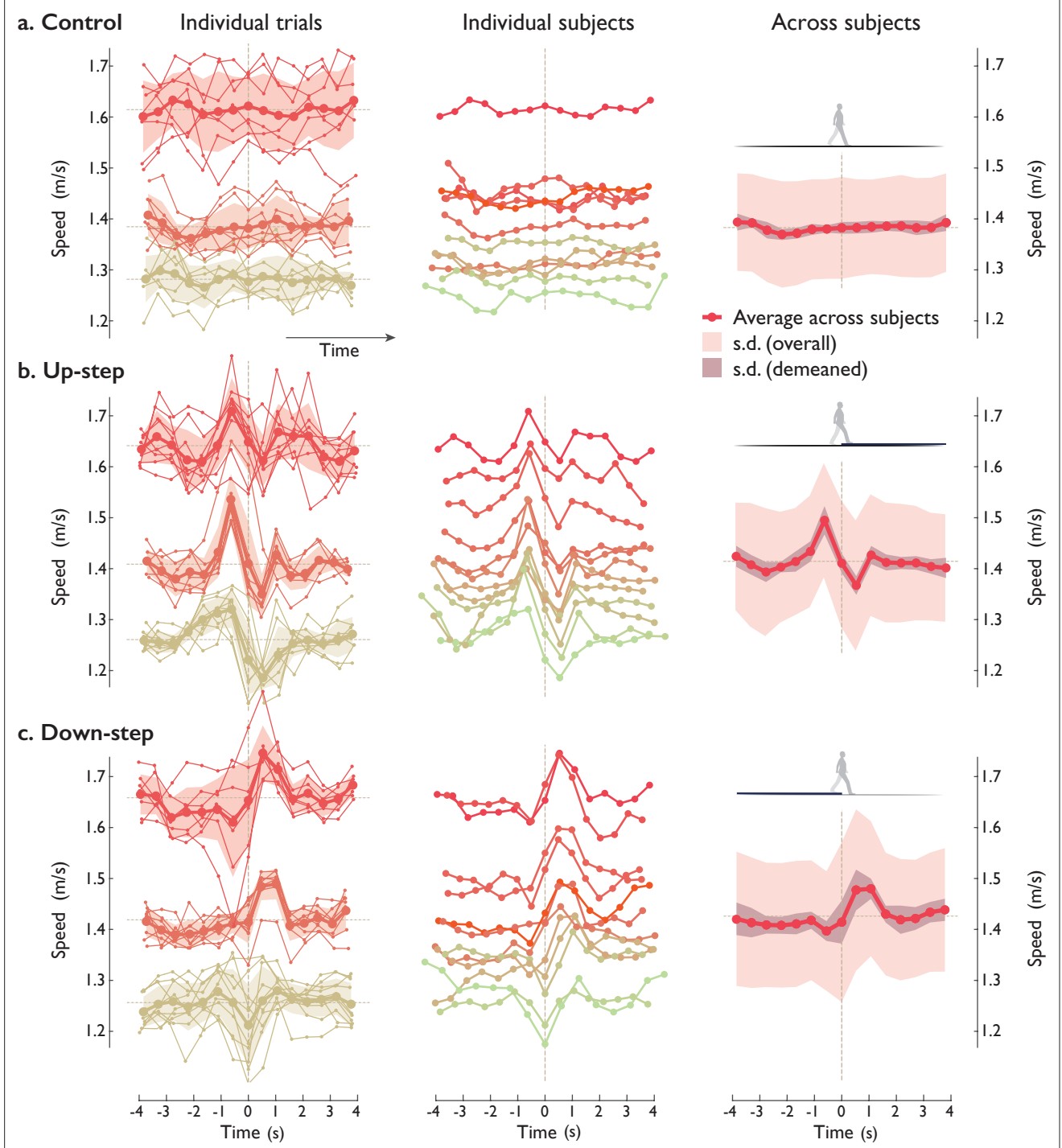

**Figure 4.** Human walking speed trajectories vs. time, for (**a**) Control and (**b**) Up- and (**c**) Down-step conditions.
Plots are arranged in columns: (left) all individual trials of three representative test subjects (thin lines connecting small dots), along with per-subject average trajectories (across trials, thick lines) and standard deviations (SDs, shaded regions ± 1 s.d.; dashed line indicates average speed). Averages were computed for speed fluctuations treated as sequences of discrete steps, and plotted in both discrete (dot symbols) and continuous time. (Middle:) Speed fluctuations of each test subject (n = 2, individual colors) walking over the step, averaging all trials within each subject. (Right:) Average speed trajectories across all test subjects (solid line), with standard deviation (SD) across all subjects (light shaded region), and SD ignoring subject-dependent speed (darker shaded region). Individual patterns (Middle) were significantly correlated with the average pattern across subjects (Right). Speed is defined as each step's length divided by corresponding step time, assigned to the middle- stance instant of each step (indicated by discrete dot symbols). All trials are aligned to zero time, defined as middle -stance instant after landing on the Up- or Down-step (both 7.5 cm high).

*Figure 4 continued*

The online version of this article includes the following source data for figure 4:

**Source data 1.** Individual and average test subject speeds plotted in *Figure 4*.

The consistency of responses across subjects was quantified as follows. The Up-step speed fluctuations were similar across subjects (compare *Figure 4*, middle and right columns), with a positive correlation coefficient between each individual's Up-step trials and the average across subjects ($\rho$ = 0.82 ± 0.1252; p = 1.26e−10 paired *t*-test of correlations, performed across discrete speed sequences). The same was true for Down-step patterns, with positive correlation ($\rho$ = 0.68 ± 0.27, p = 3.0e−06). Neither Up- nor Down-step speed fluctuations were well correlated with Control ($\rho$ = −0.016 ± 0.21 and 0.184 ± 0.193, respectively, correlating each subject's Up- and Down-step responses against average Control across subjects, p = 0.78, p = 0.007). Moreover, the speed fluctuations for the Up-step and Down-step were somewhat opposite to each other, with a negative correlation between individual Up-steps and average Down-step pattern and vice versa (respectively, $\rho$ = −0.34 ± 0.16, p = 1.6e−05; $\rho$ = −0.27 ± 0.22, p = 0.0016).

## Human compensations were consistent with minimum-work predictions

In support of the main hypothesis, the compensation strategies agreed reasonably well with optimal control model predictions (*Figure 5*, top for model; bottom for human). The model had predicted a similar triphasic pattern for speed fluctuations (*Figure 3*), as a means of traversing the walkway with minimum push-off work, while maintaining nominal overall speed. This agreement was quantified by a positive correlation coefficient between human and model fluctuations for both Up- and Down-steps ($\rho$ = 0.50 ± 0.21, p = 4.4e−6 and $\rho$ = 0.59 ± 0.17, p = 1.08e−7; paired *t*-tests for correlations performed across discrete speed sequences). And in keeping with the model's prediction of opposing fluctuations for Up- vs. Down-steps, there was also a negative correlation between human Up-steps and model Down-steps, and vice versa ($\rho$ = −0.42 ± 0.21, p = 2.75e−5 and $\rho$ = −0.54 ± 0.15, p = 8.05e−8; paired *t*-tests). We also verified that human control responses were not correlated with model predictions for either Up- or Down-step, with correlation coefficients not significantly different from zero (p = 0.32, p = 0.31).

The human triphasic responses were not consistent with the alternative control strategies. The anticipatory speed-up before perturbation did not agree with the no-compensation and reactive-compensation (*Figure 1b, c*) strategies, where no change in speed would be expected. The speed fluctuations before and after perturbation were also nearly opposite to, and therefore not consistent with the tight time regulation strategy (*Figure 1d*).

## Human compensation patterns were self-similar and scalable for step height and overall walking speed

There were three indicators of self-similarity and scaling, all consistent with model predictions. First, the significant intersubject correlations are indicative of self-similarity among different individuals within each condition. The model had predicted a similar triphasic response pattern that could be scaled in amplitude and time for different average walking speeds and step lengths (*Figure 3d*). The significant correlations (reported above, compare *Figure 4*, middle and right columns, p = 1.26e−10 for Up-steps, p = 3.0e−06 for Down-steps) show that each individual and each trial's speed fluctuations were proportional to the average pattern across subjects. Fluctuations were thus similar across a range of self-selected speeds and step lengths, with timing compared at discrete step numbers account for continuous time scaling. Thus, a single, self-similar pattern for Up- and Down-steps (*Figure 4* average across subjects, right column) was sufficient to summarize responses across individuals walking at their own pace.

Second, the speed fluctuations exhibited scaling with step height. There was a significant linear relationship between step height $b$ (+0.075, −0.075, and 0 cm) and each trial's fluctuation amplitude, with a positive linear coefficient $c_b$ (0.849 ± 0.047, mean ± c.i. 95% confidence interval, p = 9.7e−175), and a small offset ($d_b$ 0.104 ± 0.041). The coefficient indicates that Up- and Down-step responses (*Figure 4*, Human) were approximately opposite to each other, consistent with the model's predicted twofold symmetry. Not surprisingly, level control responses were approximately intermediate between

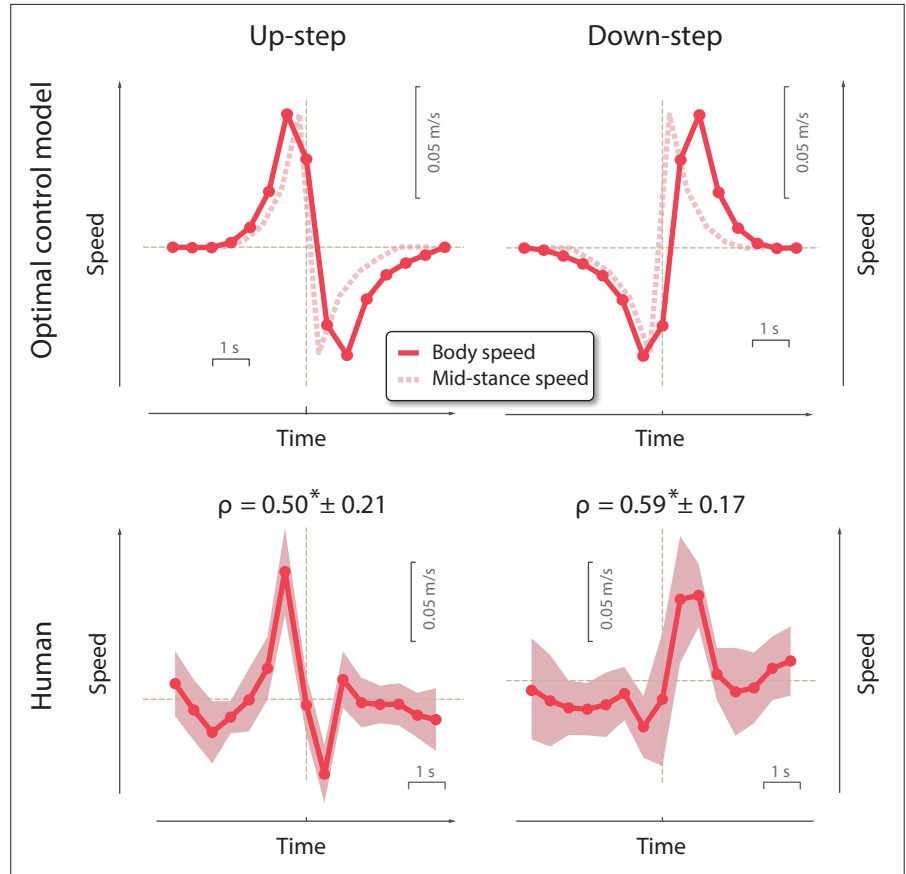

**Figure 5.** Comparison of model and human walking speed fluctuations vs. time, compensating for (left column:) Up- and (right column:) Down-steps. (Top row:) Model speed fluctuations predicted to minimize push-off mechanical work. (Bottom row:) Experimentally measured compensation strategies for humans (n = 12), showing average speed pattern across subjects (shaded regions denote ±1 s.d. after eliminating variations in average speed). Human speed fluctuations were significantly correlated with model for both Up-step ($p = 0.50 * 0.21; P = 4.4e - 6$) and Down-step ($p = 0.59 * 0.17; P = 1.1e - 7$), asterisk indicates statistical significance; correlations performed between discrete speed sequences. Each data point corresponds to the body's forward walking speed, defined as each step's length divided by step time, and assigned to the instance when the stance leg is upright. The first step onto the Up- or Down-step is indicated by vertical dashed line, also at middle stance instant. The average overall walking speed is denoted by horizontal solid line. Model trajectories (*Figure 3*) are converted from mid-stance speed in simulation (dashed lines) into the equivalent of experimental body speed (solid lines; see *Figure 6*). Model predicts shape of fluctuation pattern but not exact amplitude. Plots show dimensional equivalents to model with speed in units of $\sqrt{gL} = 3.13$ m/s and time in units $\sqrt{L/g} = 0.32$ s (using gravitational acceleration and human leg length $L = 1$ m).

the two. The compensatory fluctuation pattern for Up-steps can therefore predict a significant portion of the patterns for the other two step height conditions.

Third, the speed fluctuation amplitudes decreased with faster walking speeds. There was a significant linear relationship between each Up-step trial's average walking speed and its fluctuation amplitude. This yielded a weak but significant negative linear coefficient $c_v$ ($-0.785 \pm 0.618$ c.i., p = 0.013), with offset near unity ($d_v$, $1.008 \pm 0.073$ c.i.). This is equivalent to a 10% increment in overall speed being accompanied by a 11.1% reduction in speed fluctuation amplitude for an Up-step (1.4 m/s). The average Up-step fluctuation pattern (across all subjects) was a significant predictor of patterns for different overall speeds and two other step height conditions.

## Humans walking durations were approximately conserved despite Up- and Down-step disturbances

Subjects were approximately compliant with the instruction to maintain a similar overall speed and walking duration across trials, whether or not there was an uneven step. There were no significant differences in overall speed, overall step length, or overall duration across trials, due to experimental condition (p = 0.65, p = 0.78, and p = 0.96, respectively, repeated measures ANOVA). Overall speeds were also fairly consistent across trials within Up- or Down-step conditions (2%–3% c.v.).

The apparent conservation of walking duration contrasts with what would be expected for a no-compensation strategy (*Figure 1b*). The model, if performing constant push-offs instead of compensating, would slow-down atop the Up-step (*Figure 1b*, speed), mainly due to the exchange of kinetic energy for potential energy. It would gradually accumulate a loss of about 0.7 s in time (*Figure 1b*, time) compared to level walking. Alternatively, a particle sliding on frictionless ground at human-like speed, would be expected to lose about 0.7 m/s and 8 s to an upward ramp of equivalent height. Thus, both a walking model and a sliding particle lose substantial speed and time due to a change in height, if there were no active time conservation. Human subjects did not lose significant time from the Up-step's potential energy, nor did they gain significant time from the Down-step.

## Discussion

We examined how humans anticipate and compensate for a step change in the height of an otherwise flat walking surface. The compensatory response was characterized by a systematic, triphasic pattern in walking speed fluctuations, from which we draw several notable observations. First, the response was a scalable pattern that exhibited self-similarity, in that the same basic pattern could describe behavior at a variety of average walking speeds and step lengths. In addition, the response also exhibited an anticipatory component, meaning that it partially occurred prior to physically encountering the step. Finally, the response was consistent with predictions from a simple walking model, optimizing for least mechanical work. We next discuss these findings and their implications for anticipatory human control.

Human speed fluctuations exhibited a systematic, scalable pattern. A single basic pattern could describe the Up- and Down-step responses of different individuals, and even the different trials of a single individuals. That pattern could simply be scaled in amplitude and time scale to match any single response (similar to model, *Figure 3d*), according to the trial's average walking speed and step length (*Figure 4*). Systematicity is expected in part because of the deterministic and scalable dynamics of walking, with a pendulum-like exchange of kinetic and gravitational potential energy within each step (*Kuo, 2002*), and particularly an exchange of forward speed for vertical height atop the Up-step ($i = 0$, *Figure 2*). A pendulum's motion, and the action of the step-to-step transition, are consistent and scalable in speed and step length, and in amplitude and time (*Kuo, 2002*). The scalability could potentially make it unnecessary for the central nervous system (CNS) to synthesize or reoptimize a new compensation for each new perturbation encountered. The dynamics of walking appear consistent enough to facilitate prediction of the compensation patterns .

The observed active control pattern was anticipatory, reactive, and time conserving. The evidence for anticipation is that the speed fluctuations commenced a few steps before the perturbation, as the first phase of the triphasic pattern (e.g., steps $i = -2, -1$ in *Figure 4*), consisting of a speed-up prior to Up-steps, and a slow-down prior to Down-steps. This anticipation was not consistent with the alternative hypothetical strategies of no compensation (*Figure 1b*), reactive and nonanticipatory compensation (*Figure 1c*), and tight regulation to avoid step time fluctuations (*Figure 1d*). Rather it was consistent with the minimum-work optimal control model, and counteracted the subsequent effects of the perturbation, which were a substantial loss (gain) of speed and time atop the Up- (Down-) step, during the second phase of the triphasic pattern ($i = 0, 1$). Following that perturbation and despite the anticipation, walking speed was still slower (faster) than nominal, and thus counteracted by another gradual speed-up (slow-down) during the third, reactive phase ($i = 2, 3, \ldots$). It is unclear whether the reactive phase was produced by feedback or by planning, but its effects were apparently part of an overall compensatory plan, because all three phases together conserved overall walking duration. Nominal walking speed and nominal time were regained several steps after the perturbation. In principle, it should also be possible to compensate with a purely reactive control after the perturbation

(e.g., *Figure 1c*), an entirely anticipatory control, or an infinite number of other variations. However, model simulations suggest that such actions would require more mechanical work (and in many cases, much more) than the more integrative, triphasic pattern extending both before and after the perturbation (*Darici et al., 2020*).

This compensatory strategy suggests that humans can consider energy and time in their control decisions. Optimal control is helpful for resolving redundancies and indeterminacies, but it is not straightforward to determine an objective function that adequately represents human behaviors. Metabolic energy expenditure is an important determinant of level locomotion (*Alexander, 1996*; *Kuo et al., 2005*), and transient perturbations only add to that expenditure. We used a simple proxy measure for energy expenditure, the push-off work for step-to-step transitions. That work is proportionate to and appears to account for a majority of energetic cost of steady walking (*Adamczyk et al., 2006*; *Donelan et al., 2001*; *Donelan et al., 2002*; *Kuo et al., 2005*), and explains much of the cost of transient accelerations (*Seethapathi and Srinivasan, 2015*). The work proxy is thus predictive of physiological energy expenditure, while having minimal need for fitted parameters. Our study also explicitly considers timing, which is not a concern in steady locomotion at fixed speed. Here, considerable time (*Figure 1b*) and/or economy (*Figure 1c*) would be lost to an uncompensated perturbation (*Darici et al., 2018*; *Darici et al., 2020*), making their interaction important. We therefore incorporated overall time as a constraint in the model, and as a task instruction in the experiment. Timing is certainly important for tasks such as catching a ball, and perhaps for locomotion tasks such as intercepting a sidewalk curb or catching up to a friend on the sidewalk. It is of course a matter of context whether a person wishes to conserve time, hurry, or dawdle. But saving time often costs more energy, and our results suggest that humans can plan economical locomotion strategies that avoid loss of time when appropriate.

There remains the question of how the control is implemented by the central nervous system. The human's ability to reason about surface perturbations could be regarded as a mapping from visual terrain image and body state into a control action, equivalent to an inverse internal model of dynamics (*Kawato, 1999*). Examples from reinforcement learning suggest that such a mapping could be optimized iteratively (*Heess et al., 2017*), which might be representable with biological neurons. Our results suggest that the control policy could conceivably stored in quite compact form, because a single compensation pattern (e.g., $\Delta v^{up}$) might simply be scaled for different walking speeds, step heights, or step lengths. It is also conceivable that such a pattern could be repeatedly applied or superimposed to predict long sequences of uneven terrain, which are also energetically costly (*Kowalsky et al., 2021*; *Voloshina et al., 2013*). Also needed for learning is a means to evaluate the objective cost function, perhaps with physiological sensors of metabolic cost (*Selinger et al., 2015*), but it might be sufficient to use sensors indirectly related to motion, and learn their association with energy. There is presently little known about how the CNS actually implements locomotor compensations, but the patterns observed here are relatively simple and scalable.

This study highlights a less-appreciated aspect of vision-based path planning. It is clear that humans use vision to plan paths for the body (*Arechavaleta et al., 2008*; *Brown et al., 2021*), including adjustment of COM height (*Müller et al., 2012*) and foot placement (*Matthis and Fajen, 2013*; *Patla, 1998*). After all, it is sensible to predict which paths are feasible and which are less arduous. But we found that humans plan not just positions, but also dynamical trajectories such as forward momentum. They make quick, dynamically sensible decisions to overcome quite minor obstacles, apparently for energetic benefit. Such planning might also explain the leg loading preceding a Down-step in human running (*Müller et al., 2012*). It is also consistent with how birds run over an obstacle, with an anticipatory vault in the step beforehand ($i = -1$), perhaps for economy (*Birn-Jeffery et al., 2014*). Path planning may therefore be for more than just body location, but also dynamical state.

There are of course, other possible optimality criteria not considered here. We have tested and rejected a few here: no compensation, reactive compensation, and tight speed regulation (*Figure 1*). Other alternatives could potentially be tested, if stated unambiguously and objectively enough to model. A challenge is that most quantitative models to date apply to steady-state conditions, and it is unknown whether they can predict transients. To our knowledge, the present study is the first to use a mechanistic model to predict behavioral, multistep trajectories performed by humans. Perhaps the closest analog is a model based on empirical energetic cost curves that explain the curvilinear paths that humans take to go through doorways (*Brown et al., 2021*). We suspect that the empirical costs

could be predicted mechanistically by a three-dimensional version of our dynamic walking model (e.g., *Rebula et al., 2017*), although that remains to be tested. We also expect that our model is compatible with other, more complex models, if the center of mass (COM) moves in an inverted pendulum motion, and the mechanical work of step-to-step transitions accounts for much or most of the energy expenditure. It would be challenging to predict the observed human responses with a model that does not subscribe to these basic principles.

Such a mechanistic approach has similarities to the separate field of neuromotor control for upper extremity reaching movements. That field has demonstrated how humans learn and adapt their arm dynamics, consistent with CNS internal models (*Franklin et al., 2008*; *Sharp et al., 2011*). In reaching studies, adaptation is usually guided by repeated practice and explicit feedback, say of position error with respect to an explicit target. Trajectory planning may similarly apply to locomotion, in anticipation of perturbations to dynamics. But one difference here is that subjects received no explicit feedback of any kind of error, including timing, and no repeated practice. Their planning appears to be based on prior experience with similar dynamics and similar timing context. There may be rich opportunity for locomotion to borrow from the optimization approaches of reaching, while also providing a complementary perspective on CNS control.

The present study is a drastic simplification of the costs relevant to human locomotion. There are certainly many contributions to the metabolic cost, such as for muscles to produce force (*Rebula and Kuo, 2015*; *van der Zee and Kuo, 2021*) irrespective of work. There are also many actions known to cost energy during locomotion, for example transient adjustment of step length and frequency (*Ojeda et al., 2015*; *Snaterse et al., 2011*), active lifting of the swing foot for ground clearance (*Wu and Kuo, 2016*), and lateral step width and foot placement (*Donelan et al., 2001*; *O'Connor et al., 2012*). There are also nonenergetic factors that govern human behavior such as vigor (*Shadmehr et al., 2019*), which also affects timing, but in the form of a reward rather than the constraint used here. Stability may also be important for human locomotion, although we found no need to include it as an explicit objective, first because our walking model has passive dynamic stability against small disturbances (*Kuo, 2002*), and second because optimization for economy alone was sufficient to traverse the perturbation without falling. Stability might affect walking more when there is a more significant risk of falling than observed here. The present study shows that energy and time are two factors that may influence locomotion planning, but is also not intended to exclude other objectives and influences.

There are also limitations to the present model of walking. Notably, our model was intended to illustrate the effects of relatively small speed variations and surface height perturbations on pendulum-like walking. We did not explicitly control for walking speed, resulting in quite small differences between trials that make the speed scaling a rather weak test. We examined relatively low, 7.5 cm steps that can be traversed quite easily and still be treated as walking, as opposed to climbing stairs. The model did have predictive value (correlating with human with $\rho$ = 0.50, *Figure 4*), but the human patterns seemed to have some qualitatively consistent differences, for example in the amount of slow-down preceding the Down-step, and the number of steps spent speeding up or slowing down (*Figure 4*). The differences might be related to how humans flex their knees when stepping down, and actively lift the foot to clear the trailing step, especially if the foot lands far behind it and must therefore travel a longer distance before moving to the lower height. This complex action could potentially influence the strategy for COM motion. It could potentially be included in dynamic walking models with explicit knee (e.g., *Dean and Kuo, 2009*) and ankle (*Zelik et al., 2014*) joints, which have step-to-step transition energetics similar to here (e.g., *Donelan et al., 2002*; *Adamczyk et al., 2006*). Such models could potentially be useful for stepping higher, but we also believe that steps as high as stairsteps would ultimately require quite different control, less like an inverted pendulum. We also expect that transiently varying step lengths and foot placements could be added to the model (*Bhounsule, 2014*; *Kuo, 2001*; *Ojeda et al., 2015*). We have previously proposed that substantial energy is also expended for active leg motions during steady walking (*Doke and Kuo, 2007*; *Doke et al., 2005*), perhaps half that expended for step-to-step transitions (*Kuo, 2001*). We presently lack a model for the cost of transient step adjustments, which could potentially provide more detail about how active speed and step length fluctuations are achieved (e.g. *Appendix 2—figure 1*). It is also likely that humans couple their sagittal and frontal plane motions for a change in step height, which might be accommodated in a three-dimensional model (*Kim and Collins, 2017*; *Kuo, 1999*). Considerable improvements might

be achievable with additional dynamical degrees of freedom, but these would also entail additional free parameters, which must often be fit to data and could diminish the generality of the model predictions. An advantage of the present simple model is that it has negligible parametric sensitivity (*Figure 3d*), and predicts fundamental aspects of walking dynamics based on first principles.

Despite these limitations, we showed here that humans perform compensatory, multistep speed adjustments on uneven terrain. A simple model minimizing the mechanical work of step-to-step transitions can predict these adjustments. The adjustments start several steps before, extend after the perturbation, in a triphasic pattern of speed fluctuations. Such a pattern is consistent with metabolic energy expenditure as a criterion for optimal control, and shows that humans perform anticipatory control before a perturbation is directly encountered. The CNS appears to anticipate the effects of disturbances on the dynamics of the body and exploit these dynamics for active and economical control.

# Materials and methods

This study consisted of an experiment to test how humans traverse a single Up- or Down-step interrupting steady walking. We hypothesized that humans would anticipate the surface height perturbation, and produce a compensatory pattern of walking speed fluctuations, similar to predictions from our previous modeling study (*Darici et al., 2020*). Specifically, the model predicted a triphasic pattern for speed fluctuations, intended to compensate for the perturbation with minimal mechanical work, and conserve overall walking speed and walking time over a walkway of fixed distance. Although the model is far simpler than human, its mechanical work appears to explain a substantial fraction of human metabolic cost, perhaps enough to predict anticipatory walking strategies. We first summarize the model predictions, prior to describing the experimental procedure.

## Model of walking

We summarize predictions from an optimal control model of walking (*Figure 2*; *Darici et al., 2020*), with details in Appendix 1. The task is to walk down a walkway interrupted by a single Up- or Down-step (numbered step $i = 0$; *Figure 1a*), with adjustments to the forward speed $v_i$ of the COM for each step  (*Figure 2a*; defined as the COM velocity at midstance when stance leg is vertical). The model has rigid, pendulum-like legs supporting a point-mass pelvis of mass $M$, along with infinitesimal-mass feet (*Figure 2b*; *Kuo, 2002*). The dynamics of the single stance phase are those of a simple inverted pendulum, which conserves mechanical energy and therefore loses speed stepping on an Up-step. As a simplification, the model's legs nominally take steps either of fixed length similar to the 'rimless wheel' model (*McGeer, 1990*), or variable length increasing with speed according to the human preferred step length relationship (see Appendix 1). With inverted pendulum dynamics, a level, nominal step (*Figure 2c*) is energetically conservative, except when punctuated by a step-to-step transition, where the trailing leg pushes off (PO) impulsively just before the leading leg's dissipative collision (CO) impulse. This redirects the center-of-mass (COM) velocity to a new pendular arc described by the leading leg. The push-off and collision impulses are performed along the axis of the corresponding legs, with push-off as the only powered actuation, and (perfectly inelastic) collision the only dissipation. The push-off work constitutes the only energy cost in this model. Experiments show that it explains how mechanical work and human metabolic energy expenditure increase as a function of step length (*Donelan et al., 2002*) or step width (in 3D model; *Donelan et al., 2001*) on level ground. Of course, there are certainly other costs such as to move the legs (*Doke et al., 2005*) or maintain balance (*Donelan et al., 2004*; *O'Connor et al., 2012*), but evidence suggests that they are dominated by the higher cost of step-to-step transitions (*Kuo et al., 2005*). Here, we modeled uneven terrain as a small, vertical height discrepancy $b$ in step height, where additional push-off can help compensate for momentum lost to an Up-step (*Figure 2d*), and collision for momentum gained to a Down-step (*Figure 2e*).

Optimal control was used to determine the compensation for a single step height change (*Darici et al., 2020*). The objective was to minimize total push-off work for multiple (variable $N$) steps while compensating for the terrain unevenness. The model was governed by the walking dynamics and constrained to start and end its compensation at steady, nominal speed, with the $N$ steps distributed equally before and after the Up-step. It was also constrained to match the total time for nominal level,

steady walking, thus making up for time lost to the Up-step. The continuous time pendulum dynamics were controlled by discrete push-off work $u_i$ for each step (where $i = 0$ for the push-off onto the Up-step), which served as the decision variables for optimization. Each push-off caused a change in the forward speed $v_i$ of the COM each step (*Figure 2a*), discretely sampled at the midstance instance when the stance leg passed through vertical, prior to the step-to-step transition. The control strategy refers to the discrete sequence of speeds $v_i$ (including over a range of steps), or equivalently the push-off sequence ${}_iu_i$. We expected that the compensation would occur within several steps before and after the perturbation, and selected $N = 15$ as more than sufficient to encompass the relevant speed fluctuations. In model, velocity perturbations decay within several steps, based on the exponential persistence time of model (about 90% decay within 5.7 steps; *Darici et al., 2020*). We simulated a nominal step height $b$ equivalent to 7.5 cm.

## Human subject experiment

To test model predictions, we measured speed fluctuations as humans walked down a level walkway (about 30 m) with a single, raised step onto a second level of 7.5 cm (*Figure 1a*). We tested healthy adult subjects ($n = 12$; 7 male, 5 female, all under 30 years age), whose steps and walking speed were measured with inertial measurement units (IMUs) on both feet. There were three conditions: Up-step, Down-step, and Control on level ground. Both Up- and Down-steps used the same walkway except in opposite directions, and Control took place on level floor directly alongside the walkway. The raised section, commencing about halfway down, was assembled from fairly rigid, polystyrene insulation foam. In all conditions, subjects walked at comfortable speed from a start line through and past a finish line. Trials took place in alternating direction, with a brief delay for the subject to turn around and stand briefly before starting the next trial. There were at least 4 (and up to 10) trials of each condition, usually with Up- and Down-steps alternating with each other, except with occasional Control conditions inserted at random and interrupting that pattern. Before data collection, subjects were given opportunity to try the conditions and gain familiarity with the walkway and the location of the Up-step. For brevity, all mentions of the Up-step apply equally to the Down-step, unless explicitly stated. We tested one-sample group, with sample size ($n = 12$) selected for a statistical power of at least 0.8, based on previous data for anticipated means and SDs (*Darici, 2015*). The experiment was performed once, and each test subject was tested once with each condition as a repeated measure. All subjects provided written informed consent prior to the experiment, according to Institutional Review Board procedures (University of Michigan, Energetics, Balance, and Control of Human Locomotion, HUM00020554).

The experiment was minimally governed, aside from instructing subjects which conditions to perform. To establish a subjective 'normal' walking speed, and walking time, subjects first performed two to four Control trials at the beginning of the experiment. We provided context by instructing subjects to walk 'in about the same time' throughout the experiment, although they never received feedback about their timing, reported in Results. The instructions were to avoid behaviors that might otherwise be considered normal in daily life, for example stopping to inspect the Up-step or to tie a shoelace or deciding to hurry through the experiment. We consider a loose time goal to be more explicit about expectations than a simpler but potentially ambiguous instruction to 'walk normally'. At the same time, we allowed for arbitrary self-selected speeds, because the model's predictions do not depend on a particular speed. We thus expected a range of speeds across trials and across subjects. To help subjects to step onto the Up-step without stutter steps, there was a visual cue (a paper sticker) on the otherwise featureless walking surface, several steps (approximately 5 m) before the Up-step. Subjects were informed that they could use this cue to line up their steps prior to the Up-step, although they were not required to use it, and no trials were excluded even if there was a stutter step. Anecdotally, most subjects appeared to pay little attention to the sticker, especially after the first few trials.

We measured walking speed fluctuation trajectories (*Figure 6*) with IMUs. A discrete body speed was recorded for each walking step, based on an integrated trajectory for each foot (*Rebula et al., 2013*). The trajectories were computed from an IMU mounted atop each foot (*Figure 6a*), with gravity-compensated accelerations integrated to yield a spatiotemporal trajectory, subject to an assumption that each foot comes briefly to rest during each footfall, approximately in the middle of stance when the foot is flat on ground. These instances were used to correct the integrated foot velocity to zero,

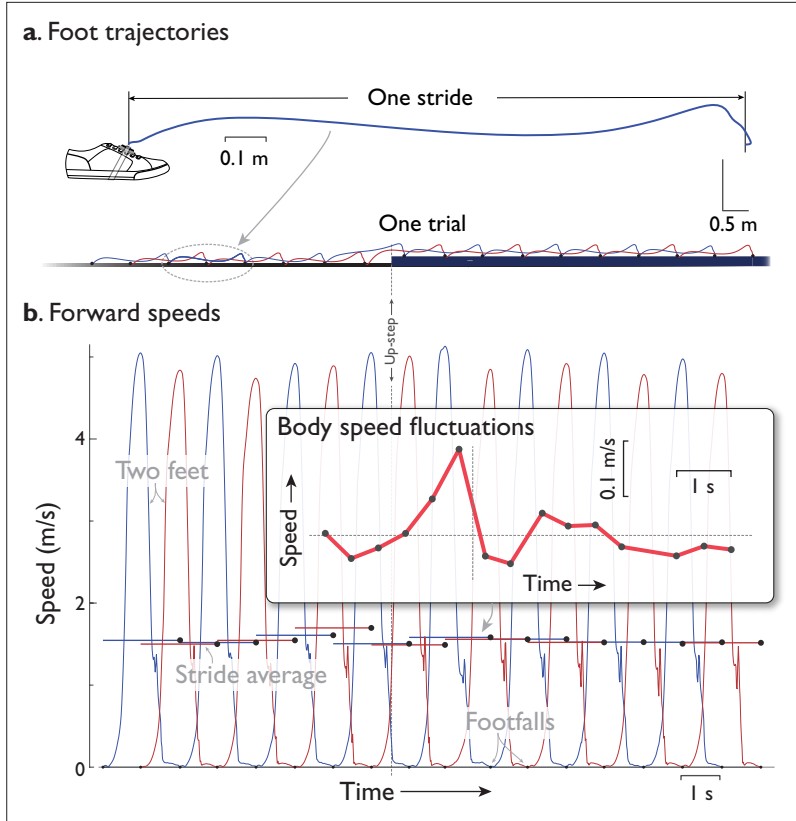

**Figure 6.** Experimental measurement of forward walking speed from inertial data (representative data). (**a**) Foot trajectories in space are computed from inertial measurement unit (IMU) data. Shown are representative sagittal plane trajectory for one foot's motion during a stride, and the individual strides for both feet during an Up-step trial (with 2× vertical scale for trial). Trajectories were integrated from an IMU atop each foot, where drift was reduced by detecting footfall instances (black dots) where the foot was stationary, and correcting the integrated velocity to zero at those instances (**Rebula et al., 2013**). (**b**) Forward speed vs. time for each foot and for the body. Each foot's instantaneous forward speed was computed from the spatial trajectories. The average speed during each stride ('stride average', filled symbols) was determined, equal to the distance traveled divided by the stride duration. This was then treated as the body speed for each step (inset figure, filled symbols are same as stride averages), sampled discretely from each foot's strides at the respective footfall instances. The fluctuations in body speed were compared with model predictions.

and thus reduce integration drift (**Rebula et al., 2013**). We then estimated stride length and time from the forward distance and time between an IMU's footfalls, respectively. Individual distances traveled by the 2 feet (**Figure 6b**) were corrected for integration drift so that they both agreed on overall distance, using linear detrending. Each foot's average speed over a stride (**Figure 6c**) was defined as the stride length between two discrete footfall (midstance) instants of the same foot, divided by the stride time between those instants. The body's forward walking speed (Speed, **Figure 6c** and subsequent figures) was estimated as a discrete sequence of these average speeds, once per step by alternating left and right feet (assuming the body travels at the same distance per stride as the feet), with each sample $i$ corresponding to the midstance instance at beginning of stride. These speed trajectories were analyzed for a central, 8.5 m segment of the walkway, or about 15 steps surrounding the Up-step (**Figure 1a**). This was more than sufficient to encompass the compensation patterns, which largely took place within only a few steps before and after the perturbation. To compare between trials, the time $t = 0$ was defined as the instant of the footfall onto the Up-step (or Down-step, or step next to it for Control), as detected by IMU. The speed trajectory for each subject's trials within a condition was averaged at discrete step numbers, as were the times for those steps, to yield an individual's average speed trajectory per condition.

Model predictions were tested with speed trajectories for Up-step, Down-step, and Control conditions. First, we tested the consistency of human responses across different speeds and different subjects. Each individual's trajectories (as discrete sequences) were compared against the average trajectory for all individuals with Pearson's correlation coefficient $\rho$, with significant correlation determined by one-sample $t$-test. The patterns were thus compared for their shape for each discrete step, and not their amplitude or time scale. Significant correlation was therefore also used as an indicator of self-similarity across speeds, step lengths, and subjects.

Next, as a test of the model (**Figure 3**), individual subject trajectories were correlated against the model's predicted trajectories. The model was intended to predict proportionalities in speed and work, and therefore the shapes of the fluctuation patterns, rather than absolute quantities. Therefore, each individual's speed fluctuation pattern of $N$ steps was correlated against the predicted pattern, with the Pearson's correlation coefficient $\rho$ expected to be positive (as opposed to a null hypothesis value of zero). Positive correlation across all subjects was tested for significance with one-sample paired $t$-test. The model predicted similar fluctuation patterns regardless of parameter choices (**Figure 3d**), and so we compared human against the model with fixed step lengths for simplicity. Only the optimal control model was tested in this way, whereas the alternative strategies (**Figure 1**) were tested by inspection.

Finally, we tested the model's predicted scaling of speed fluctuations with step height and speed. We expected that the average Up-step response could predict the amplitude each individual's response to Down-step and level control and across different speeds. These were tested with two types of linear regression, with the hypothesized predictor $\Delta v_i^{up}$ denoting the average Up-step speed fluctuation pattern (across all subjects and Up-step trials, for steps $i = -(N-1)/2$ to $(N-1)/2$, with $\Delta$ denoting fluctuation from average). The fluctuation waveform $\Delta v^{jk}$ for a subject $j$ and trial $k$ was expected to scale in proportion to the trial's step height $b^{jk}$ (+7.5, 0, and −7.5 cm),

$$\text{step height scaling}: \quad \Delta v_i^{jk} \approx \left( c_b \frac{b^{jk}}{b^{up}} + d_b \right) \Delta v_i^{up} \quad \text{for all steps } i$$

where $b^{up}$ is the Up-step height, and $c_b$ and $d_b$ the coefficient and offset from regression. Similarly, the fluctuation waveform was expected to scale in negative proportion to a trial's overall speed $v^{jk}$,

$$\text{Speed scaling}: \quad \Delta v_i^{jk} \approx \left( c_v \frac{\bar{v}^{jk}}{\bar{v}^{up}} + d_v \right) \Delta v_i^{up} \quad \text{for all steps } i$$

where $c_v$ and $d_v$ are speed-related coefficient and offset, and $v^{up}$ the average walking speed for all Up-step trials. Scaling with step height was tested by statistical significance of $c_b$, expected to be near unity. Scaling with speed was tested by significance of $c_v$, expected to be slightly negative. The offsets $d_b$ and $d_v$ were expected to be near zero and unity, respectively, but were not tested because only $c_b$ and $c_v$ were central to the hypothesis. We did not explicitly control for walking speed, and so the speed test relies on a sufficient range of differences in self-selected speeds between trials. Statistical significance was tested with a p threshold of 0.05.

## Acknowledgements

This work is supported by NSF DGE 0718128, the ONR ETOWL program, NIH AG030815, the Dr. Benno Nigg Research Chair (University of Calgary), and NSERC (Natural Sciences and Engineering Research Council of Canada) Discovery program and Canada Research Chair (Tier 1) program.

## Additional information

### Funding

| Funder | Grant reference number | Author |
|---|---|---|
| National Science Foundation | | Osman Darici<br>Arthur D Kuo |
| Office of Naval Research | ETOWL program | Osman Darici<br>Arthur D Kuo |

| Funder | Grant reference number | Author |
| --- | --- | --- |
| National Institutes of Health | AG030815 | Osman Darici<br>Arthur D Kuo |
| University of Calgary | | Osman Darici<br>Arthur D Kuo |
| Natural Sciences and Engineering Research Council of Canada | Discovery program | Osman Darici<br>Arthur D Kuo |
| Canada Research Chairs | | Osman Darici<br>Arthur D Kuo |

The funders had no role in study design, data collection, and interpretation, or the decision to submit the work for publication.

## Author contributions
Osman Darici, Formal analysis, Investigation, Writing - original draft, Writing - review and editing; Arthur D Kuo, Funding acquisition, Investigation, Supervision, Writing - original draft, Writing - review and editing

## Author ORCIDs
Osman Darici (iD) http://orcid.org/0000-0001-6217-5656

## Ethics
All subjects provided written informed consent prior to the experiment, according to Institutional Review Board procedures (University of Michigan, Energetics, Balance, and Control of Human Locomotion, HUM00020554).

## Decision letter and Author response
Decision letter https://doi.org/10.7554/eLife.65402.sa1
Author response https://doi.org/10.7554/eLife.65402.sa2

# Additional files

## Supplementary files
• Transparent reporting form

## Data availability
All data generated or analyzed during this study are included in the manuscript and supporting file.

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

# Appendix 1

## Dynamic walking model

The model dynamics are briefly summarized as follows (detailed previously by *Darici et al., 2018*). Each of $N$ steps has index with the Up- or Down-step disturbance located at $i = 0$ (*Figure 1c*). Negative therefore refer to the preparatory steps beforehand, and positive to recovery steps thereafter. Each step has a pendulum-like single stance phase with passive dynamics, and a costly step-to-step transition. Mechanical work is only performed during that transition, starting with COM velocity $v_i^-$ directed forward and downward at the end of each stance phase. For brevity, the equations presented here use dimensionless versions of quantities, with $M$, $g$, and $L$ as base units. The step-to-step transition starts with pre-emptive push-off work $u_i$ (in units of mass-normalized work) performed impulsively along the trailing leg to redirect the COM velocity. This is followed immediately by the heel-strike collision along the leading leg, to yield postcollision velocity $v_i^+$. Again applying impulse momentum (*Kuo, 2002*),

$$v_i^+ = v_i^- \cos 2\alpha + \sqrt{2u_i} \sin 2\alpha. \tag{1}$$

where $2\alpha$ is a fixed interleg angle (*Figure 2*).

Another single stance phase follows the step-to-step transition, and is modeled as an underactuated, simple inverted pendulum. As a discrete indicator of overall forward momentum, we use the midstance velocity $v_i$ (no superscript; see *Figure 1*) following step-to-step transition $v_i^-$, sampled when the leg is vertical, and the COM velocity is purely forward.

We treat steady, level walking as the nominal condition (*Figure 1c*). The nominal push-off work $u_i$ offsets the collision work (*Kuo, 2002*), so that

$$u_i = \tfrac{1}{2}(v_i^-)^2 \tan^2 \alpha \tag{2}$$

and $v_i^+ = v_i^-$. The uneven step disturbs steady walking (*Figure 1d*). Its height $b$ (positive for Up-steps, negative for Down-steps) causes the preceding stance phase to end with a different stance leg angle from nominal. For a given height $b$ and step length $S$, we define the angular disturbance as $\delta_i$,

$$\delta_0 = \sin^{-1} \tfrac{b}{S}, \delta_i = 0 \text{ for } i \neq 0 \tag{3}$$

where the angle is zero for all nondisturbance steps.

An inverted pendulum stance phase follows each step-to-step transition. A step time $\tau_i$ defined as the time for the stance leg angle $\theta$ to move between successive step-to-step transitions, from $v_i^+$ to $v_{i+1}^+$ and passing through midstance speed $v_i$.

Using the linearized dynamics, the dimensionless step time $\tau_i$ of step is

$$\tau_i = \log \frac{\alpha - \delta_{i+1} \sqrt{(v_i^+)^2 - 2\alpha(\delta_i + \delta_{i+1}) + \delta_{i=1}^2 - \delta_i^2}}{v_i^+ - \alpha - \delta_i}. \tag{4}$$

Solving the equation of motion with the step time, the velocity at end of stance $v_{i+1}^-$, or equivalently the beginning of the next step-to-step transition can be found as:

$$v_{i+1}^- = \tfrac{1}{2}(e^{-\tau_i}(v_i^+ + \alpha + \delta_i) + e^{\tau_i}(v_i^+ - \alpha - \delta_i)). \tag{5}$$

Midstance time $\tau_i'$ for step can also be found using the linearized dynamics:

$$\tau_i' = \log(\frac{\sqrt{(v_i^+)^2 - \alpha^2 - 2\alpha\delta_i - \delta_i^2}}{v_i^+ - \alpha - \delta_i}) \tag{6}$$

Solving for mid-stance speed $v_i$,

$$v_i = \tfrac{1}{2}\left(e^{-\tau_i'}(v_i^+ + \alpha + \delta_i) + e^{\tau_i'}(v_i^+ - \alpha - \delta_i)\right) \tag{7}$$

We chose nominal parameters to correspond to typical human walking, with two ways to specify step length. The nominal gait was for a person with leg length $L$ of 1 m walking at 1.5 m/s, with step length of 0.79 m and step time of 0.53 s (from anecdotal observations). This step length was kept

fixed in the optimization, and parameter sensitivities were computed for a longer fixed value (0.95 m in *Figure 3d*), detailed more extensively previously (*Darici et al., 2020*). The other specification was to increase step length according to the preferred human relationship, increasing approximately with $v^{0.42}$ (*Grieve, 1968*) and thereby changing the work cost slightly. Optimization yielded similar shaped speed fluctuation patterns with little sensitivity to fixed or variable step length (*Figure 3d*). In model, the sole optimization cost was push-off work, neglecting costs for moving the legs back and forth (*Kuo, 2001*), expected to be fairly constant for the small speed fluctuations predicted by model.

Using dynamic similarity, parameters and results may be expressed in terms of body mass $M$, gravitational acceleration $g$, and $L$ as base units. The corresponding model parameters treated as dimensional are angle $\alpha = 0.41$, push-off $U = 0.0342MgL$, step time $T = 1.665g^{-0.5}L^{0.5}$, and precollision speed $V = 0.601g^{0.5}L^{0.5}$, where capital letters indicate nominal values for $u_i$, $\tau_i$, and $v_i^-$, respectively. We also refer to a nominal speed $V = 0.44g^{0.5}L^{0.5}$ for midstance speed $v_i$. We considered a range of Up-step heights, for example $b = 0.075L$, equivalent to about 7.5 cm for a human.

## Optimization problem

The optimization is formulated as follows, with policy $\pi$ denoting the set of push-offs $u_i$:

$$\pi = \underset{\pi}{argmin} \sum_{i=-(N-1)/2}^{(N-1)/2} u_i$$

subject to:

Speed: $v_{-(N-1)/2} = V, v_{(N-1)/2} = V$

Time: $\sum_{i=-(N-1)/2}^{(N-1)/2} \tau_i = T \cdot N$

Dynamics: Model dynamics (above)

where $N$ is the total (odd) number of steps and step $i = 0$ is the first step on the Up-/Down-step. Thus, $N$ adjusts how far in advance and or after the perturbation for which the model can modulate its momentum or speed. The speed constraints are such that the initial and final conditions are equal to the nominal, steady speed $V$. The time constraint makes up for lost time, so that the total time is equal to the nominal time to walk $N$ steps on level ground. By the end of the control sequence, the model must walk at the same speed as nominal and must have caught up with the nominal model on level ground. We chose $N$ large enough to cover the speed adjustments that humans made in the experiments. Note that, because human speeds are most conveniently measured from footfall to footfall, we converted the model speeds to a similar footfall definition (stride length divided by stride time, footfall to footfall) for purposes of comparison between model and human (*Figure 5*). The optimization problem was formulated and solved using a constrained optimization solver (Matlab 'fmincon', MathWorks, Inc, Natick, MA, USA).

## Appendix 2

### Step duration and step length fluctuations

Reported here are human subject compensation patterns for Up- and Down-step perturbations (*Appendix 2—figure 1*), in terms of step durations and step lengths. This study was primarily concerned with the fluctuations in forward speed or momentum that humans perform to compensate for a step height perturbation. Humans can modulate their speed with transient changes to both step length and step time, a feature not included in the present model. The human data are included here to facilitate future models that could provide greater detail on these responses.

We did not perform hypothesis tests on these data, but a few descriptive measures are provided. For Up-steps (*Appendix 2—figure 1* left/top row), step lengths were about +3.1%, +18.5%, and −7% for the three steps surrounding the perturbation ($i = −1, 0, + 1$), respectively, compared to average step length. For Down-steps, the corresponding step length differences were 0.24%, 2.29%, and −2.58% (*Appendix 2—figure 1*, bottom row). As for step times (*Appendix 2—figure 1*, bottom), the Up-step differences were −5.8%, +15%, and −2.7%, and Down-steps differences were +2.5%, +2.5%, and −3.5%, compared to average step period. Qualitatively, humans appeared to distribute compensations differently between Up- and Down-steps. Whereas speed fluctuations were approximately opposite in amplitude (*Figure 4*) for Up- vs. Down-steps, the same does not appear true for step durations (*Appendix 2—figure 1*, left column) and step lengths (*Appendix 2—figure 1*, right column). We speculate that the human knee affords different strategies for regulating forward momentum when stepping upward vs. downward.

a. Up-step

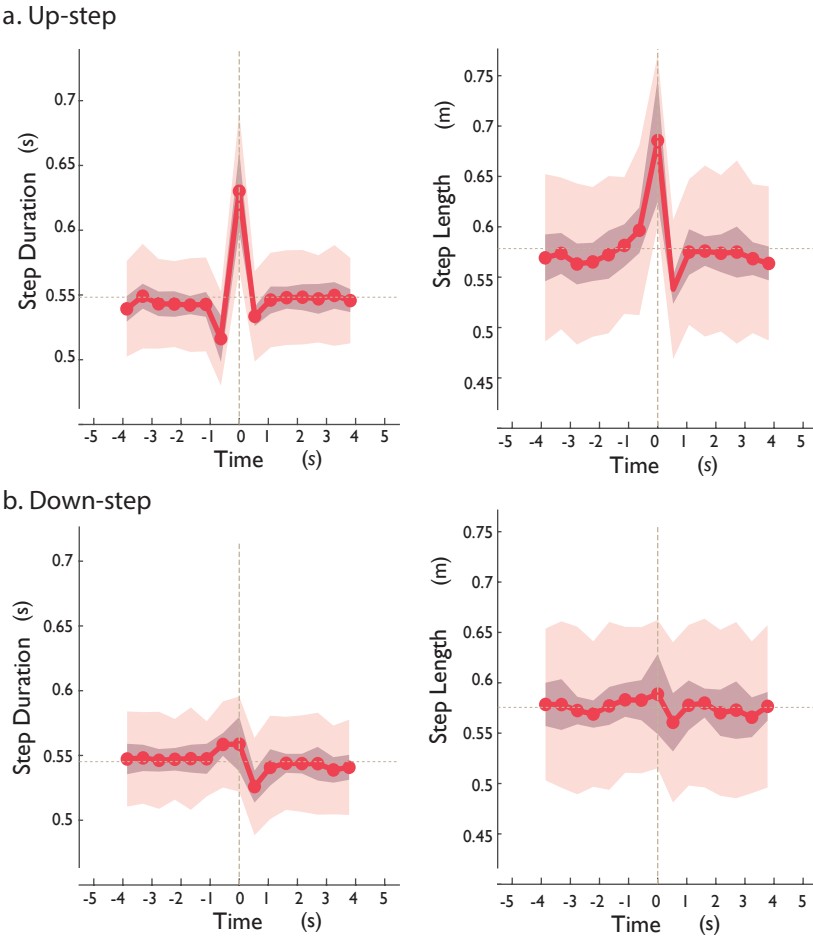

**Appendix 2—figure 1.** Human (left column) empirical step duration and (right column) step length fluctuations vs. time, for (**a**) Up-step and (**b**) Down-step. Shown are step durations and step lengths for each step (line denotes mean across subjects, shaded area denotes ±1 SD; *N* = 12) vs. time, with vertical line denoting the step onto

*Appendix 2—figure 1 continued*

the perturbation. The human speed fluctuations (**Figure 4**) appeared to be achieved by transiently changing a combination of step durations and step lengths. Each discrete speed is defined as each step's corresponding step length divided by duration.

These observations suggest possibilities for incorporation into a model. To predict transiently changing step durations and lengths, the model would need an associated energetic cost. The only cost included here is for step-to-step transitions, which appear to account for the majority of human energy expenditure during walking (**Kuo et al., 2005**). We have proposed that humans pay an additional cost for moving the legs back and forth, which could explain the energetically optimal step length for steady walking (**Kuo, 2001**), and could account for about one-third of its overall cost (**Doke et al., 2005**). Transient step adjustments would be expected to alter or add to the steady cost, in amount not yet modeled, because the mechanistic determinants remain unknown. Nevertheless, the existence of such a cost does not exclude step-to-step transition costs as an important determinant of the speed fluctuations observed here.

