## [Editor Report]

The study combines theoretical and experimental approaches to probe the laws governing strategy for coping with the control of stepping on uneven terrain. Congruent results in anticipatory and reactive adjustments indicate that a simple strategy based on the conservation of energy may be expressed within neural control pathways for locomotion.

---

## [Decision Letter]

**Decision letter after peer review:**

Thank you for submitting your article "Humans optimally anticipate and compensate for an uneven step during walking" for consideration by *eLife*. Your article has been reviewed by 3 peer reviewers, and the evaluation has been overseen by a Reviewing Editor and Aleksandra Walczak as the Senior Editor. The following individual involved in review of your submission has agreed to reveal their identity: Manoj Srinivasan (Reviewer #3).

Essential revisions:

1) Stride length: The authors need to address whether the model limitations of using a fixed stride length affect the generalization of the results to variable stride lengths. The generalization of the claims, even in a qualitative way, need to be demonstrated, particularly as the authors have previously published simulation predictions with variable stride length. The reviewers agree that even a qualitative match between simulations and experiments would be sufficient given the inherent limitations of a simple model.

2) Timing: Several questions regarding the claims with respect to timing require clarification and potentially further simulations. The time constraint imposed on the simulation is a particular concern; the total time should be released or at a minimum, the cost of different total times should be compared. Without doing so, the claim that there is no time lost due to the perturbation are not supported. Another issue with the claims around timing are the potential bias based on instruction to the participants. These multiple issues need to be addressed to substantiate the claims around timing.

3) Given that the simulations appear to have been published previously, the contributions of this manuscript above and beyond those of previous ones by the authors need to be more fully described. What is the major advance that this paper provides?

4) Aspects of the study methodology require clarification. These include the tested hypotheses, subject instructions and the scientific rationale for the study design and outcomes. Considerations for alternative study design and study limitations should be addressed in the discussion.

5) A more complete discussion of the modeling assumptions and limitations needs to be included. These should include both the limitation of a simple model, the effects of including a swing limb, and alternative optimality criteria that may provide either similar or disparate results.

*Reviewer #1 (Recommendations for the authors):*

1) The suggestion that the automatic optimization strategy is engaged here does not indicate whether this solution is dictated by subject instructions. In other words, could the solution here be simply the fact that subjects would be instructed not to stop when transversing the obstacles? The implied automatic action for this adaptation is not fully supported by experiments.

2) The description of background information could benefit from a more thorough review of the related studies and clarifications.

3) The study refers to the feedforward limb speed control in the study design and outcomes; yet, no background for the CNS using this parameter as a variable is provided. If this is an assumption then it should be identified as such.

4) One of the main findings is the self-similarity of the anticipatory response at multiple average speeds and step lengths. This brings a few concerns related to the normalization (see specific question 6) and the overall significance of this outcome. The discussion should identify the impact of this finding on the conclusions.

5) The overall study design appears to be unclear. The interplay between the tested hypotheses and the background or theoretical facts is not fully clear. The understanding of study validity could be improved by the additional description of study design and the consideration of alternatives and study limitations. For example, stepping over obstacles could be executed with multiple strategies and probably in a context-dependent way. These alternatives and the experimental observations should be discussed in the context of the findings in this study.

6) P2, ln8 "This raises the question.…" The general point expressed here may benefit from the focus on the specific question. As is, the outcome may be task-dependent, e.g., the interception of prey would require a different strategy from that used to catch a ball.

7) P2, ln22 "The negotiation of uneven terrain.…" The background literature highlighting these points could be added. The predictive planning in the same context was previously demonstrated in studies by Patla in humans and in animals by Drew.

8) p2, ln43 "… humans compensate for ground perturbations is unknown…" This may be an overgeneralization. There are multiple types of ground perturbations. Some mechanisms are known and documented, e.g., the mechanisms of the stretch reflex and finite step phase transition rules in humans and animals (including insects), e.g., Prochazka's models.

9) p3, ln23 "… swing phase dynamics are ignored, and the legs are constrained to fixed step lengths.…" This oversimplification does not agree with the observed experimental results. The dependency of limb speed on the control of the swing phase has a direct bearing on the assumption of limb speed control. The implications of this limitation should be clarified. Ultimately, the addition of swing dynamics with the appropriate stride length dependency could substantially support the claims in this study.

10) p4, ln10 "… discretely sampled a the mid-stance instance.…" The measurement of step speed should be clarified. Have you estimated the speed of the center of mass or some other related kinematic landmark?

11) Figure 2 The analysis is done in time even though the speed of each subject was self-paced. The previous studies indicate that humans may use discrete in time step-based control, which would imply that the analysis in Figure 2 should be similarly normalized to each step.

12) p6, ln25 "… systematic behavior.…" This statement may be unclear. What is the nature of these systematic variations amongst volunteers?

13) p6, ln28 " Humans compensated for.…" This statement makes a specific claim based on the presented testing; however, the hypothesis for the tests has not been expressed. Please, consider clarifying the nature of the study design here. Was this work driven by the rationale or the specific a priori hypotheses?

14) p6, ln33 "… would lose 1 s.…" This statement likely implies a full stop for a second prior to the step over the obstacle. Please confirm or clarify the origin of this conclusion. Also, it is unclear if this simulated behavior depended on the initial conditions. For example, the step adjustments may only be necessary if the timing with the obstacle interaction is not good enough for the smooth step over it.

15) p7, ln9 "The Up-step and.…" Please, clarify what variables are correlated here. It is unclear how the "conditions" could be correlated.

16) Methodological unclarity. What was the placement of IMU units? Was there any signal processing of IMU signals? The instrumentation of subjects should be described in methods.

*Reviewer #2 (Recommendations for the authors):*

The conclusions of the paper could be strengthened by:

1) demonstrating that the predictions also hold for a model with variable step length. It is not clear why step length was fixed as a fixed step length does not seem a valid assumption here and previous models of the authors had variable step length. Does a model with variable step length predict variations in step length in agreement with experimental observations?

2) demonstrating that optimal control also predicts no time lost due to stepping up as compared to walking on even terrain. If I got it right, the total time is imposed when solving the optimal control problem. The authors stress that time is lost in uncompensated up-steps when describing their model predictions, which they contrast with the optimal policy. However, the loss of time was not predicted but imposed (page 14 – line 27).

3) demonstrating that the number of compensation steps N that minimizes work corresponds to the observed number of compensation steps. Now, N is informed by observations (page 14 – line 29). N is not specified, I infer from the figures that it was nine but according to the author's previous modeling paper, there is little additional reduction in work when increasing the number of compensation steps above three. Therefore, minimal work does not explain why people take more than three compensation steps.

4) demonstrating that minimal work leads to better agreement between simulations and observations than other plausible optimality criteria. I understand that some plausible criteria, e.g., stability against noise, are not easy to test but they could be discussed.

5) demonstrating that the predicted dependence of speed fluctuations on step height is in agreement with experimental observations. (I realize that this might require additional experiments and might therefore not be feasible within a reasonable time but I feel it is a missed opportunity that this was not part of the original experimental protocol.)

*Reviewer #3 (Recommendations for the authors):*

The manuscript is well written. I found it easy to read.

[Editors' note: further revisions were suggested prior to acceptance, as described below.]

Thank you for resubmitting your work entitled "Humans optimally anticipate and compensate for an uneven step during walking" for further consideration by *eLife*. Your revised article has been reviewed by 3 peer reviewers and the evaluation has been overseen by a Reviewing Editor and by Aleksandra Walczak as the Senior Editor.

*Reviewer #1 (Recommendations for the authors):*

The thorough revision is appropriate for publication.

*Reviewer #2 (Recommendations for the authors):*

The authors have addressed some but not all the questions/concerns raised in the previous review.

Previous essential revision 1. Instead of demonstrating that minimization of mechanical work captured observed stride length variability, the authors moved the results on stride length variability to the appendix. I am under the impression that the authors have previously published simulations in which stride times were not imposed (was not contradicted in response) and it is unclear why simulations with variable (i.e. not pre-imposed) stride times could not be performed. The added value of performing simulations with different imposed stride lengths is limited as compared to the added value of performing simulations in which stride length is a free variable (and thus follows from optimization). Simulations with variable stride lengths could have given information on whether the key assumption (minimization of mechanical work) predicts other features than speed fluctuations alone.

Previous essential revision 5. The authors have introduced alternative hypotheses but this feels 'after the facts'. It was already clear that these hypotheses would not hold (with prior knowledge of the experimental data) and they have therefore limited value. We specifically asked about alternative optimality criteria and such criteria will only strengthen the reported findings when they truly challenge them.

With respect to stability as an alternative optimization criterion, I find the addition to the discussion not convincing. The model being able to traverse the perturbation without falling tells little about stability against noise, which can be expected in real life (but was not modeled). I am also unsure about what the authors mean by 'the model could easily traverse the perturbation'. How was 'easily' defined?

*Reviewer #3 (Recommendations for the authors):*

The authors have addressed all my specific comments satisfactorily.

I am also reasonably satisfied with how the authors have addressed the issue of constant vs varying step length that other reviewers raised and the consensus review included. However, the authors may want to clearly note that allowing the step length to vary with speed via a static preferred speed-step-length relationship is not the same (ie not mathematically equivalent) as allowing the step length to vary as an independent optimization variable because, well, one problem has more optimization degrees of freedom than the other. But this additional calculation is certainly suggestive (and I believe its suggestion) that allowing step lengths to vary will not change the speed fluctuation results substantially.

---

## [Author Response]

Essential revisions:1) Stride length: The authors need to address whether the model limitations of using a fixed stride length affect the generalization of the results to variable stride lengths. The generalization of the claims, even in a qualitative way, need to be demonstrated, particularly as the authors have previously published simulation predictions with variable stride length. The reviewers agree that even a qualitative match between simulations and experiments would be sufficient given the inherent limitations of a simple model.

We have added more justification that fixed stride lengths are not critical to model predictions. We added predictions using the human preferred stride length vs. speed relationship, and show that the optimal speed fluctuations are quite similar whether using fixed or preferred stride lengths (Figure 5d; Methods text; more detail in Appendix). This is in addition to the existing predictions for different fixed stride lengths and different average speeds.

2) Timing: Several questions regarding the claims with respect to timing require clarification and potentially further simulations. The time constraint imposed on the simulation is a particular concern; the total time should be released or at a minimum, the cost of different total times should be compared. Without doing so, the claim that there is no time lost due to the perturbation are not supported. Another issue with the claims around timing are the potential bias based on instruction to the participants. These multiple issues need to be addressed to substantiate the claims around timing.

We have made several changes to address the question of timing. As suggested, we have added a simulation with the total time is released (Figure 1b), showing how time is lost to the Up-step (also shown previously, Darici et al., 2020). We have also clarified that the (unreleased) model is constrained to conserve time, and that the prediction is that time can be conserved if desired, not that it must necessarily be so. Regarding the experiment, we have clarified how participants were instructed to loosely maintain similar time (Methods), with the resulting conclusion that they were able to approximately conserve, not that they always do. In fact, the instruction was given because humans who walk entirely without context might spontaneously decide to slow down, speed up, pause, or stop at any time. We were not clear that conserved time was not a hypothesis, and now express it more explicitly as an observation. The corresponding Result subheading is “Humans walking durations were approximately conserved despite Up- and Down-step disturbances.” Finally, we have included model predicted costs for the alternative strategies vs. optimal (Model predictions).

3) Given that the simulations appear to have been published previously, the contributions of this manuscript above and beyond those of previous ones by the authors need to be more fully described. What is the major advance that this paper provides?

We have taken a number of measures throughout the manuscript to emphasize how the present study is a human subjects experiment, and the previous was a model simulation paper. For example, an added figure (new Figure 6) shows more information about the experimental measurements. We have re-emphasized the experiment in a number of places in the text, for example:

Abstract: “Here we show that humans compensate…” and “In experiment, humans…”

Introduction: “The purpose of the present study was to test whether humans negotiate a step height disturbance with an anticipatory compensation strategy as predicted by dynamic optimization.”

Methods: “This study consisted of an experiment…”

Results: “1. Humans produced…”, “2. Human compensations…”, “3. Humans…were self-similar”, “4. Human walking durations…” [emphasis on Humans, as opposed to previous Model paper]

Discussion: “We examined how humans anticipate…”

4) Aspects of the study methodology require clarification. These include the tested hypotheses, subject instructions and the scientific rationale for the study design and outcomes. Considerations for alternative study design and study limitations should be addressed in the discussion.

A number of changes have been made regarding methodology. These are detailed in the responses below, but here is a brief summary of some of the changes:

Introduction: Expanded rationale, e.g. “The results may reveal whether humans reason about their walking dynamics to perform predictive planning on uneven terrain.”

Methods, Model: “The model yielded two main predictions to be tested experimentally.” [three paragraphs including alternative hypotheses]

Figure 1 and text: Added alternative hypotheses for no compensation, non-anticipatory compensation, and tight regulation to avoid speed fluctuations.

Human subjects experiment: “Model predictions were tested with speed trajectories.” [three paragraphs describing tests]. Also expanded description of subject instructions.

Discussion: Rejection of alternative hypothesis, e.g. “This anticipation was not consistent with the alternative hypothetical strategies…”

5) A more complete discussion of the modeling assumptions and limitations needs to be included. These should include both the limitation of a simple model, the effects of including a swing limb, and alternative optimality criteria that may provide either similar or disparate results.

We have added alternative hypotheses (Figure 1b, c, d) in comparison with results. We have also expanded the discussion of model limitations, including about the swing limb and particularly its knee. These points are detailed in the responses below.

Reviewer #1 (Recommendations for the authors):1) The suggestion that the automatic optimization strategy is engaged here does not indicate whether this solution is dictated by subject instructions. In other words, could the solution here be simply the fact that subjects would be instructed not to stop when transversing the obstacles? The implied automatic action for this adaptation is not fully supported by experiments.

We have clarified that the evidence for optimality is the observed tri-phasic pattern, and not the conservation of walking duration. As reviewer, suggests, duration was a matter of subject instructions, and we merely verified that they followed instructions. We have reorganized the results to clarify the emphasis on the pattern as hypothesis test (Results first three sub-headings), and we have de-emphasized walking duration (Results, last sub-heading “Human walking durations were approximately conserved.”).

2) The description of background information could benefit from a more thorough review of the related studies and clarifications.

We have added more context to the background information (specific points 2, 3), to clarify what aspects of uneven terrain are already understood, and how the present study addresses the prediction of compensation strategies with dynamic optimization.

3) The study refers to the feedforward limb speed control in the study design and outcomes; yet, no background for the CNS using this parameter as a variable is provided. If this is an assumption then it should be identified as such.

We have clarified that speed was intended to refer to the body’s forward speed. We had been unclear that we used inertial measurements to measure speed of foot to infer the average speed of the body with each step. We have revised the Methods to clarify that the hypothesis and experimental measurements are concerned with fluctuations in walking speed for the whole body. A new Figure 6 shows how the body’s forward speed is estimated from foot speeds.

4) One of the main findings is the self-similarity of the anticipatory response at multiple average speeds and step lengths. This brings a few concerns related to the normalization (see specific question 6) and the overall significance of this outcome. The discussion should identify the impact of this finding on the conclusions.

We have expanded on the discussion of self-similar scaling. Early in Discussion:

“A pendulum’s motion, and the action of the step-to-step transition, are consistent and scalable in speed and step length, and in amplitude and time (Kuo, 2002). The scalability could potentially make it unnecessary for the central nervous system (CNS) to synthesize or re-optimize a new compensation for each new perturbation encountered.”

Later in Discussion of control:

“the control policy could conceivably be stored in quite a compact form, because a single compensation pattern (e.g., Δvup) might simply be scaled for different walking speeds, step heights, or step lengths (Figure 4). It is also conceivable that such a pattern could be repeatedly applied or superimposed to predict long sequences of uneven terrain, which are also energetically costly (Voloshina et al., 2013; Kowalsky et al., 2021).”

5) The overall study design appears to be unclear. The interplay between the tested hypotheses and the background or theoretical facts is not fully clear. The understanding of study validity could be improved by the additional description of study design and the consideration of alternatives and study limitations. For example, stepping over obstacles could be executed with multiple strategies and probably in a context-dependent way. These alternatives and the experimental observations should be discussed in the context of the findings in this study.

We have re-organized the Methods to improve the link between model and experiment. We have expanded on alternative hypotheses, as summarized in the above replies, and in responses below. More extensive details are included in the responses below.

6) P2, ln8 “This raises the question.…” The general point expressed here may benefit from the focus on the specific question. As is, the outcome may be task-dependent, e.g., the interception of prey would require a different strategy from that used to catch a ball.

We have reworded the sentence to focus on the differences with the curb:

“A sidewalk curb may similarly be anticipated ahead of time, albeit with most of the dynamics within the person rather than the curb, and with a less clearly defined objective.”

7) P2, ln22 "The negotiation of uneven terrain.…" The background literature highlighting these points could be added. The predictive planning in the same context was previously demonstrated in studies by Patla in humans and in animals by Drew.

We added reference to Drew, and moved the reference to Patla early within the paragraph. We have also expanded the background on planning based on other references:

“In particular, foot placement and motion are planned, with the help of vision, to avoid or step over upcoming obstacles (Patla, 1998; Patla and Rietdyk, 1993), and to land on foot targets (Drew et al., 2008; Matthis and Fajen, 2013). Although much of this planning has been described in terms of kinematics, it may also include intersegmental dynamics, which may be regulated to clear obstacles with reduced mechanical work (Patla and Prentice, 1995).”

8) p2, ln43 "… humans compensate for ground perturbations is unknown…" This may be an overgeneralization. There are multiple types of ground perturbations. Some mechanisms are known and documented, e.g., the mechanisms of the stretch reflex and finite step phase transition rules in humans and animals (including insects), e.g., Prochazka's models.

We have reworded to be more specific about our hypothesis about anticipatory strategies:

“This raises the question of whether humans anticipate the curb, and what objective criteria govern its interception.” We did not intend to implicate stretch reflexes and other effects.

9) p3, ln23 "… swing phase dynamics are ignored, and the legs are constrained to fixed step lengths.…" This oversimplification does not agree with the observed experimental results. The dependency of limb speed on the control of the swing phase has a direct bearing on the assumption of limb speed control. The implications of this limitation should be clarified. Ultimately, the addition of swing dynamics with the appropriate stride length dependency could substantially support the claims in this study.

We had intended to focus on speed of the body, and not on swing phase. We have made several changes to de-emphasize swing phase. We have also added model results regarding stride length dependency:

“As a simplification, the model’s legs nominally take steps either of fixed length similar to the “rimless wheel” model (McGeer, 1990), or variable length increasing with speed according to the human preferred step length relationship (see Appendix 1)”.

In experimental methods we have clarified how we measured body speed, and added a figure to illustrate (Figure 6). We also moved the stride length and timing results to Appendix, since these were only measurements without any hypothesis tests. This may help focus on the main hypothesis, where body speed was compared between model and human (Figure 6).

10) p4, ln10 “… discretely sampled a the mid-stance instance.…” The measurement of step speed should be clarified. Have you estimated the speed of the center of mass or some other related kinematic landmark?

We added clarification that the model’s speed is for center of mass:

“forward speed vi of the COM each step i(Figure 2a; defined as the COM velocity at mid-stance when stance leg is vertical).”

In the later Experiment sub-section, we define human body speed as the average body speed for each step, to compare against model.

11) Figure 2 The analysis is done in time even though the speed of each subject was self-paced. The previous studies indicate that humans may use discrete in time step-based control, which would imply that the analysis in Figure 2 should be similarly normalized to each step.

The reviewer is correct that the dynamics are continuous in time, and that the model and human should be compared in terms of discrete steps. This is indeed the case, and we have made several changes to clarify.

In Model, “Each push-off caused a change in the forward speed vi of the COM each step i (Figure 2a), discretely sampled at the mid-stance instance when the stance leg passed through vertical, prior to the step-to-step transition.”

And in Figure 2, “The model’s compensatory response is therefore summarized as a time-varying trajectory of speeds vi, discretely sampled once per step i.”

In Figure 3, “Averages were computed for speed fluctuations treated as sequences of discrete steps, and plotted in both discrete (dot symbols) and continuous time.”

12) p6, ln25 “… systematic behavior.…” This statement may be unclear. What is the nature of these systematic variations amongst volunteers?

This was a small degree of unexplained correlation in the control trials, and so we have clarified:

“This suggests a degree of unexpected, non-random behavior shared between subjects, of relatively small amplitude of about 0.014 m/s, and at around 17% of the non-Control amplitudes.”

13) p6, ln28 " Humans compensated for.…" This statement makes a specific claim based on the presented testing; however, the hypothesis for the tests has not been expressed. Please, consider clarifying the nature of the study design here. Was this work driven by the rationale or the specific a priori hypotheses?

We agree this was unclear, and in fact the conservation of walking duration was simply an observation, not a hypothesis. To emphasize the hypothesis tests, we have re-ordered the results so that the hypothesis tests appear first, each with a specific claim, and the observation about walking duration (now Result 4) with the observation “4. Humans walking durations were approximately conserved”.

We also re-worded to state “Subjects were approximately compliant with the instruction… There were no significant differences in overall speed…”

Here are the three claims that did include hypothesis tests (in Results):

1. Humans produced triphasic Up- and Down-step compensatory speed fluctuations. [tested by correlation between average human response and individual subject fluctuations]

2. Human compensations were consistent with minimum-work predictions. [tested by correlation between model and individual subject fluctuations]

3. Human compensation patterns were self-similar and scalable for step height and overall walking speed. (tested by correlation and linear regression with step height and speed as predictors)

14) p6, ln33 "… would lose 1 s.…" This statement likely implies a full stop for a second prior to the step over the obstacle. Please confirm or clarify the origin of this conclusion. Also, it is unclear if this simulated behavior depended on the initial conditions. For example, the step adjustments may only be necessary if the timing with the obstacle interaction is not good enough for the smooth step over it.

We have provided more information about the loss:

“The model, if performing constant push-offs instead of compensating, would slow down atop the Up-step (Figure 1b, speed), mainly due to the exchange of kinetic energy for potential energy. It would gradually accumulate a loss of about 0.7 s in time (Figure 1b, time) compared to level walking.”

We also added a sub-figure (Figure 1b) to illustrate the outcome.

15) p7, ln9 "The Up-step and.…" Please, clarify what variables are correlated here. It is unclear how the "conditions" could be correlated.

We have clarified that we were referring to the speed fluctuations:

“The Up-step speed fluctuations were similar across subjects (compare Figure 3 middle and right columns)”.

16) Methodological unclarity. What was the placement of IMU units? Was there any signal processing of IMU signals? The instrumentation of subjects should be described in methods.

We have added more detail on placement and processing in the text. Regarding placement:

“The trajectories were computed from an IMU mounted atop each foot (Figure 6a).”

Regarding signal processing:

“with gravity-compensated accelerations integrated to yield a spatiotemporal trajectory, subject to an assumption that each foot comes briefly to rest during each footfall.”

The new figure (Figure 6) was added to illustrate this.

Reviewer #2 (Recommendations for the authors):The conclusions of the paper could be strengthened by:1) demonstrating that the predictions also hold for a model with variable step length. It is not clear why step length was fixed as a fixed step length does not seem a valid assumption here and previous models of the authors had variable step length. Does a model with variable step length predict variations in step length in agreement with experimental observations?

We have included more explanation, including optimal control predictions with the model constrained to follow the preferred human step length vs. speed relationship (Figure 5d). In text:

“A basic pattern for optimal speed fluctuations retains approximately the same shape across different overall walking speeds, fixed step lengths, or even step length changing to the human preferred step length relationship.”

Later in the same paragraph:

“…we expect a single basic pattern, treated as a sequence of discrete speed fluctuations (Figure 5a) to predict optimal responses regardless of an individual’s average speed, step length, or the preferred speed and step length relationship.”

In Discussion, we also note:

“We also expect that transiently varying step lengths and foot placements could be added to the model (Bhounsule, 2014; Kuo, 2001; Ojeda et al., 2015b), and potentially provide more detail about how active speed fluctuations are achieved (e.g. Appendix 2-figure 1).”

2) demonstrating that optimal control also predicts no time lost due to stepping up as compared to walking on even terrain. If I got it right, the total time is imposed when solving the optimal control problem. The authors stress that time is lost in uncompensated up-steps when describing their model predictions, which they contrast with the optimal policy. However, the loss of time was not predicted but imposed (page 14 – line 27).

The reviewer is correct, “no time lost” is not a prediction but an imposed constraint. As clarification, we have added to Methods:

“It (the model) was also constrained to match the total time for nominal level, steady walking, thus making up for time lost to the Up-step.”

In Experiment: “We provided context by instructing subjects to walk “in about the same time” throughout the experiment.”

In Results, we have de-emphasized the time conservation, by placing it after the hypothesis tests (regarding speed fluctuations), and by stating more clearly that:

“Subjects were approximately compliant with the instruction to maintain a similar overall speed and walking duration across trials, whether or not there was an uneven step.”

3) demonstrating that the number of compensation steps N that minimizes work corresponds to the observed number of compensation steps. Now, N is informed by observations (page 14 – line 29). N is not specified, I infer from the figures that it was nine but according to the author’s previous modeling paper, there is little additional reduction in work when increasing the number of compensation steps above three. Therefore, minimal work does not explain why people take more than three compensation steps.

We have clarified that the value of N is not a hypothesis:

“We expected that the [model] compensation would occur within several steps before and after the perturbation, and selected N=15 as more than sufficient to encompass the relevant speed fluctuations.”

And in Human experiment, “These speed trajectories were analyzed for…about 15 steps surrounding the Up-step (Figure 1a). This was more than sufficient to encompass the compensation patterns, which largely took place within only a few steps before and after the perturbation.”

Reviewer is correct that compensation takes place in roughly three steps before and after perturbation, for about seven steps in total. We optimized and measured for more steps to ensure that the entire compensation was observed. In Results:

“The basic speed response… (1) Speed up in the two steps prior, (2) then lose speed during the two steps onto the Up-step and immediately thereafter, and (3) then regain speed over the following one or two steps.”

Also in Results:

“The anticipatory compensation thus begins only a few steps before the perturbation, but is accompanied by several steps of response after the perturbation.”

We do not claim an absolute minimal number of steps for human, but rather describe when and how most of the response occurs.

4) demonstrating that minimal work leads to better agreement between simulations and observations than other plausible optimality criteria. I understand that some plausible criteria, e.g., stability against noise, are not easy to test but they could be discussed.

We have expanded on the Discussion of optimality criteria, including a number of additional energy costs and objectives (e.g., no compensation, non-anticipatory reaction, and tight regulation to avoid speed changes (Figure 2)). Regarding stability, we have added the following remark:

“Stability may also be important for human locomotion, although we found no need to include it as an explicit objective because the model could easily traverse the perturbation, and found it most economical to do so without falling. Stability might affect walking more when there is a more significant risk of falling than observed here.”

5) demonstrating that the predicted dependence of speed fluctuations on step height is in agreement with experimental observations. (I realize that this might require additional experiments and might therefore not be feasible within a reasonable time but I feel it is a missed opportunity that this was not part of the original experimental protocol.)

We have expanded on the Discussion of optimality criteria, including a number of additional energy costs and objectives (e.g., no compensation, non-anticipatory reaction, and tight regulation to avoid speed changes (Figure 2)). Regarding stability, we have added the following remark:

“Stability may also be important for human locomotion, although we found no need to include it as an explicit objective because the model could easily traverse the perturbation, and found it most economical to do so without falling. Stability might affect walking more when there is a more significant risk of falling than observed here.”

[Editors' note: further revisions were suggested prior to acceptance, as described below.]

Reviewer #2 (Recommendations for the authors):The authors have addressed some but not all the questions/concerns raised in the previous review.Previous essential revision 1. Instead of demonstrating that minimization of mechanical work captured observed stride length variability, the authors moved the results on stride length variability to the appendix. I am under the impression that the authors have previously published simulations in which stride times were not imposed (was not contradicted in response) and it is unclear why simulations with variable (i.e. not pre-imposed) stride times could not be performed. The added value of performing simulations with different imposed stride lengths is limited as compared to the added value of performing simulations in which stride length is a free variable (and thus follows from optimization). Simulations with variable stride lengths could have given information on whether the key assumption (minimization of mechanical work) predicts other features than speed fluctuations alone.

We have made three changes to address this comment. First, we added a new paragraph in Model Predictions to explain why we test predictions from step-to-step transition costs. Second, we added a discussion (in Discussion) regarding transient step durations. It is true that we have previously proposed a cost for steady stride times, but we explain that the transient stride adjustments observed here require a different model, which has not been proposed. Third, we added a paragraph to Appendix expanding on this further. These changes are detailed as follows.

1. We have moved the model predictions earlier, to precede Results. (An editor had previously asked us to move Methods to the end, but we find it clearer to restore model predictions to their original location as a separate section.) We clarify:

“We hypothesize that step-to-step transitions are costly enough to merit anticipation and compensation for uneven steps. […] It is, however, not necessary to model such features, if step-to-step transitions are costly enough to predict substantial and nontrivial compensations.”

2. Added to Discussion:

“We have previously proposed that substantial energy is also expended for active leg motions during steady walking (Doke and Kuo, 2007; Doke et al., 2005), perhaps half the amount expended for step-to-step transitions (Kuo, 2001). We presently lack a model for the cost of transient step adjustments, which could potentially provide more detail about how active speed and step length fluctuations are achieved.”

3. Added to Appendix 2:

“These observations suggest possibilities for incorporation into a model. […] Nevertheless, the existence of such a cost does not exclude step-to-step transition costs as an important determinant of the speed fluctuations observed here.”

Previous essential revision 5. The authors have introduced alternative hypotheses but this feels 'after the facts'. It was already clear that these hypotheses would not hold (with prior knowledge of the experimental data) and they have therefore limited value. We specifically asked about alternative optimality criteria and such criteria will only strengthen the reported findings when they truly challenge them.

We have expanded on the Discussion of alternative hypotheses to address this comment. We understand the request (from first review), “demonstrating that minimal work leads to better agreement between simulations and observations than other plausible optimality criteria,” but it is difficult to predict what alternatives another person would accept as plausible and truly challenging. In the absence of a specific suggestion, we have added the following to the Discussion:

“There are of course, other possible optimality criteria not considered here. […] It would be challenging to predict the observed human responses with a model that does not subscribe to these basic principles.”

With respect to stability as an alternative optimization criterion, I find the addition to the discussion not convincing. The model being able to traverse the perturbation without falling tells little about stability against noise, which can be expected in real life (but was not odelled). I am also unsure about what the authors mean by ‘the model could easily traverse the perturbation’. How was ‘easily’ defined?

To clarify, we have amended the sentence (changes emphasized):

“Stability may also be important for human locomotion, although we found no need to include it as an explicit objective, first because our walking model has passive dynamic stability against small disturbances (Kuo, 2002), and second because optimization for economy alone was sufficient to traverse the perturbation without falling.”

We agree that our optimization tells little about stability against noise, but only said that “we found no need” for explicit stability, not that it does not matter at all. The rest of the paragraph already said, “Stability might affect walking more when there is a more significant risk of falling than observed here. The present study shows that energy and time are two factors that may influence locomotion planning, but is also not intended to exclude other objectives and influences.”

Reviewer #3 (Recommendations for the authors):The authors have addressed all my specific comments satisfactorily.I am also reasonably satisfied with how the authors have addressed the issue of constant vs varying step length that other reviewers raised and the consensus review included. However, the authors may want to clearly note that allowing the step length to vary with speed via a static preferred speed-step-length relationship is not the same (ie not mathematically equivalent) as allowing the step length to vary as an independent optimization variable because, well, one problem has more optimization degrees of freedom than the other. But this additional calculation is certainly suggestive (and I believe its suggestion) that allowing step lengths to vary will not change the speed fluctuation results substantially.

In Discussion, we have added:

“We presently lack a model for the cost of transient step adjustments, which could potentially provide more detail about how active speed and step length fluctuations are achieved (e.g. Appendix 2-figure 1).”

In the Appendix, we added a paragraph to expand on this issue:

“These observations suggest possibilities for incorporation into a model. […] Nevertheless, the existence of such a cost does not exclude step-to-step transition costs as an important determinant of the speed fluctuations observed here.”